# Towards Optimal Effective Resistance Estimation

**Rajat Vadiraj Dwaraknath**
Stanford University
rajatvd@stanford.edu

**Ishani Karmarkar**
Stanford University
ishanik@stanford.edu

**Aaron Sidford**
Stanford University
sidford@stanford.edu

## Abstract

We provide new algorithms and conditional hardness for the problem of estimating effective resistances in $n$-node $m$-edge undirected, expander graphs. We provide an $\widetilde{O}(m\epsilon^{-1})$-time algorithm that produces with high probability, an $\widetilde{O}(n\epsilon^{-1})$-bit sketch from which the effective resistance between any pair of nodes can be estimated, to $(1 \pm \epsilon)$-multiplicative accuracy, in $\widetilde{O}(1)$-time. Consequently, we obtain an $\widetilde{O}(m\epsilon^{-1})$-time algorithm for estimating the effective resistance of all edges in such graphs, improving (for sparse graphs) on the previous fastest runtimes of $\widetilde{O}(m\epsilon^{-3/2})$ [Chu et. al. 2018] and $\widetilde{O}(n^2\epsilon^{-1})$ [Jambulapati, Sidford 2018] for general graphs and $\widetilde{O}(m + n\epsilon^{-2})$ for expanders [Li, Sachdeva 2022]. We complement this result by showing a conditional lower bound that a broad set of algorithms for computing such estimates of the effective resistances between all pairs of nodes require $\widetilde{\Omega}(n^2\epsilon^{-1/2})$-time, improving upon the previous best such lower bound of $\widetilde{\Omega}(n^2\epsilon^{-1/13})$ [Musco et. al. 2017]. Further, we leverage the tools underlying these results to obtain improved algorithms and conditional hardness for more general problems of sketching the pseudoinverse of positive semidefinite matrices and estimating functions of their eigenvalues.

## 1 Introduction

In a weighted, undirected graph $G$ the *effective resistance* between a pair of vertices $a$ and $b$, denoted $r_G(a, b)$, is defined as the energy of a unit of electric current sent from $a$ to $b$ in the natural resistor network induced by $G$. Effective resistances arise for a broad set of graph processing tasks and have multiple equivalent definitions. For example, $r_G(a, b)$ is proportional to the expected roundtrip commute time between $a$ and $b$ in the natural random walk induced on the graph and when $\{a, b\}$ is an edge in the graph, it is proportional to the probability that the edge is in a random spanning tree.

Effective resistances also form a particular class of metrics on the vertices [1,2] and are a key measure of proximity between vertex pairs. Correspondingly, effective resistances can arise in a variety of data analysis tasks. For example, effective resistances have been used in social network analysis for measuring edge centrality in social networks [3] as well as for measuring chemical distances [4].

Effective resistances have a broad range of algorithmic implications. Sampling edges of a graph using effective resistance is known to efficiently produce cut and spectral sparsifiers (sparse graphs which approximately preserve cuts, random walk properties, and more) [5–7]. Effective resistance-based graph sparsifiers have also been applied to develop fast graph attention neural networks [8], to design graph convolutional neural networks for action recognition [9], to sample from Gaussian graphical models [10], and beyond [11,12]. Effective resistances have also been used in algorithms for maximum flow problems, [13–16, 16–18], sampling random spanning trees [19–21], and graph partitioning [22, 23]. More recently, effective resistances have also been used to analyze the problem of oversquashing in GNNs and in designing algorithms to alleviate oversquashing [24–26] and have been applied to increase expressivity when incorporated as edge features into certain GNNs [27].

37th Conference on Neural Information Processing Systems (NeurIPS 2023).

**Algorithms.** Given the broad utility of effective resistances, there have been many methods for estimating and approximately compressing them [5, 28–31]. In this paper, our main focus is the following effective resistance estimation problem. (We use $x \approx_\epsilon y$ as shorthand for $(1 - \epsilon)y \leqslant x \leqslant (1 + \epsilon)y$ and assume all edge weights in graphs are positive. See Section 2 for notation more broadly).

**Definition 1** (Effective Resistance Estimation Problem). *In the* effective resistance estimation problem *we are given an undirected, weighted graph $G = (V, E, w)$, a set of vertex pairs $S \subseteq V \times V$, and $\epsilon \in (0, 1)$ and must output $\tilde{r} \in \mathbb{R}^S$ such that with high probability (whp.), $\tilde{r}_{(a,b)} \approx_\epsilon r_G(a, b)$ for all $(a, b) \in S$.*

The state-of-the-art runtimes for solving the effective resistance estimation problem on $n$-node, $m$-edge graphs are given in Table 2. To contextualize these results, consider the special case of estimating the effective resistance of a graph's edges, i.e., when $S = E$. This special case appears in many of the aforementioned applications, e.g., [13–15, 19, 20]. The state-of-the-art runtimes for this problem are $\widetilde{O}(n^2 \epsilon^{-1})$ due to [28] and $\widetilde{O}(m \epsilon^{-1.5})$ due to [30]. A major open problem is whether improved runtimes, e.g., $\widetilde{O}(m \epsilon^{-1})$ (which would subsume prior work), are attainable.

One of the main results of this paper is resolving this open problem in the case of well-connected graphs, i.e., expanders. Expanders are a non-trivial, previously studied, special case that is often the first step or a key component for developing more general algorithms [32]. In particular we provide an $\widetilde{O}(m \epsilon^{-1} \overline{\kappa}(G))$ time algorithm where $\overline{\kappa}$ is a measure of the graph's connectivity. Previously, the only non-trivial improvement in this setting was an independently obtained runtime of $\widetilde{O}(m + n \epsilon^{-2}(\overline{\kappa}(G))^{3/2})$ for graphs with $\overline{\kappa}(G) = \widetilde{O}(1)$ [29]. We improve upon the result of [29] for sparse enough graphs and, as we explain in Section 1.1, we can almost match the result of [29] (up to an $m^{o(1)}$ factor) in dense graphs. [1]

Interestingly, we obtain our main result by providing new effective resistance sketch algorithms.

**Definition 2** (Effective Resistance Sketch). *We call a randomized algorithm an $(\mathcal{T}_s, \mathcal{T}_q, s)$-effective resistance sketch algorithm if given an input $n$-node, $m$-edge undirected, weighted graph $G = (V, E, w)$ and $\epsilon \in (0, 1)$ in time $O(\mathcal{T}_s(G, \epsilon))$ it creates a binary string of length $O(s(G, \epsilon))$ from which when queried with any $a, b \in V$, it outputs $\tilde{r}_{a,b} \approx_\epsilon r_G(a, b)$ whp. in time $O(\mathcal{T}_q(G, \epsilon))$.*

Effective resistance sketching algorithms immediately imply algorithms for the effective resistance estimation problem. We obtain our result by obtaining an $(\widetilde{O}(m \epsilon^{-1}), \widetilde{O}(1), \widetilde{O}(n \epsilon^{-1}))$-effective resistance sketch algorithm for expanders (see Section 1.1 for a comparison to prior work).

**Lower Bounds.** Given the central role of effective resistance estimation and the challenging open-problem of determining its complexity, previous work has sought complexity theoretic lower bounds for the problem. [33] showed a conditional lower bound of $\Omega(n^2 \epsilon^{-1/13})$ for the problem by showing that an algorithm that computes effective resistances in $(n^2 \epsilon^{-1/13+\delta})$ for some $\delta > 0$ time could be used to obtain a subcubic algorithm for the *triangle detection problem*, that we define below.

**Definition 3** (Triangle Detection Problem). *Given an $n$-node undirected unweighted graph $G = (V, E)$, determine whether there are distinct $a, b, c \in V$ with $\{a, b\} \in E$, $\{b, c\} \in E$ and $\{c, a\} \in E$.*

Currently, the only known subcubic algorithms for the triangle detection problem leverage fast matrix multiplication (FMM) and therefore their practical utility (in the worst case) is questionable.

**Theorem 1** (Informal, [34]). *Given an algorithm which solves the triangle detection problem in subcubic time, we can produce a subcubic algorithm, which only performs combinatorial operations and uses the triangle detection algorithm, for Boolean matrix multiplication (BMM) and additional problems which currently are not known to be solvable subcubicly without FMM.*

This theorem implies that any subcubic algorithm for triangle detection that doesn't use FMM implies a subcubic BMM algorithm that doesn't use FMM. Consequently, subcubic triangle detection is a common hardness assumption used to illustrate barriers towards improving non-FMM based methods, e.g., the effective resistance estimation algorithms of this paper.

---

[1]While our algorithms for the effective resistance estimation problem (Definition 1) were obtained indpendently, our writing and discussion of effective resistance sketch algorithms (Definition 1) was informed by their paper. We provide a more complete comparison of our work with prior results in Table 1.

In this paper we take a key step towards closing the gap between the best known running times for effective resistance estimation and lower bounds by improving the conditional lower bound of $\Omega(n^2\epsilon^{-1/13})$ to $\widetilde{\Omega}(n^2\epsilon^{-1/2})$ for randomized algorithms. We show this lower bound holds *even for expander graphs*, and hence our effective resistance estimation algorithm (as well as [28]) are optimal up to an $\epsilon^{-1/2}$-factor among non-FMM based algorithms, barring a major breakthrough in BMM.

**Broader Linear Algebraic Tools.** The effective resistance between vertex $a$ and vertex $b$ in a graph $G$ has a natural linear algebraic formulation. For all $a, b \in V$ it is known that $r_G(a, b) = \vec{\delta}_{a,b}\mathbf{L}_G^{\dagger}\vec{\delta}_{a,b}$, where $\mathbf{L}_G \in \mathbb{R}^{V \times V}$ is a natural matrix associated with $G$ known as the *Laplacian matrix* and $\vec{\delta}_{a,b} = e_a - e_b$ (see Section 2 for notation). Thus, sketching effective resistances can be viewed as problems of preserving information about subsets of entries of the pseudoinverse of a Laplacian.

Both our algorithms and lower bounds develop more general tools for handling related problems for more general (not-necessarily Laplacian) matrices. On the algorithmic side, we show our techniques can also lead to algorithms and data structures for computing certain quadratic forms involving well-conditioned SDD and PSD matrices. On the hardness side, we show our techniques also improve triangle detection hardness bounds for estimating various properties of the singular values of a matrix.

**Paper Organization.** In the remainder of the introduction we provide a more precise statement and comparison of our results in Section 1.1. In the remainder of the paper we cover preliminaries in Section 2, present upper bounds in Section 3, and present lower bounds in Section 4. Omitted proofs and additional discussion of related proofs are deferred to the supplementary material.

## 1.1 Our Results

**Algorithms.** Here we outline our main algorithmic results pertaining to effective resistance sketching and estimation, and in Section 3 we describe a extensions of our work to broader linear algebraic problems involving SDD and PSD matrices. Our main algorithmic result is a new efficient effective resistance sketch for *expanders*, a term which is used to refer to graphs with $\widetilde{\Omega}(1)$-expansion.

**Definition 4** (Expander). *For $\alpha > 0$, we say that a graph $G = (V, E, w)$ has $\alpha$-expansion if $\alpha \leqslant \phi(G)$, where $\phi(G)$ denotes the conductance of $G$ and is defined as*

$$\phi(G) := \min_{S \subseteq V, S \notni \{0, V\}} \frac{\sum_{\{u,v\}:u \in S, v \in V \backslash S} w_{u,v}}{\min\left(\sum_{u \in S} d_u, \sum_{v \in V \backslash S} d_u\right)}, \text{ where } d_u := \sum_{\{u,v\} \in E} w_u.$$

**Theorem 2.** *There is an $(\widetilde{O}(m\epsilon^{-1}), \widetilde{O}(1), \widetilde{O}(n\epsilon^{-1}))$-effective resistance sketch algorithm for graphs with $\widetilde{\Omega}(1)$-expansion.*

Table 1 summarizes and compares our Theorem 2 to previous work on effective resistance sketches, including naive algorithms to explicitly compute the pseudoinverse of the Laplacian of $G$, which can be computed in $O(n^{\omega})$ time using FMM or $\widetilde{O}(mn)$ time using a Laplacian system solver (labeled Solver). [2] A $(\mathcal{T}_s, \mathcal{T}_q, s)$ effective-resistance sketch algorithm implies an $O(\mathcal{T}_s + |S|\mathcal{T}_q)$ algorithm for the effective resistance estimation problem. Hence, Theorem 2 implies the following.

**Theorem 3** (Effective Resistance Estimation on Expanders). *There is an $\widetilde{O}(m\epsilon^{-1} + |S|)$ time algorithm which solves the effective resistance estimation problem for graphs with $\widetilde{\Omega}(1)$-expansion.*

Effective resistance sketches are a common approach to solving the effective resistance estimation problem; but there are also approaches to the problem that do not explicitly construct effective resistance sketches. Table 2 summarizes prior work on effective resistance estimation more broadly.

There has been a long line of research on the problem of computing sketches and sparsifiers of graph Laplacians [5, 28, 30, 36] (i.e., computing a sparse graph $G'$ such that quadratic forms in the Laplacian of $G'$ approximately preserves quadratic forms in the Laplacian of $G$). Building on this work, [30] showed there is an algorithm which processes a graph $G$ on $n$ nodes and $m$ edges in $O(m^{1+o(1)})$ time and produces a sparse sketch graph $H$ with only $\widetilde{O}(n\epsilon^{-1})$ edges such that $r_G(a, b) \approx r_H(a, b)$

---

[2]$\omega \leqslant n^{2.37188}$ [35] denotes the fast matrix multiplication constant.

for all $a, b \in V$. Consequently, any algorithm which runs in $\widetilde{O}(m\epsilon^{-c})$ on expanders can be improved to run in $\widetilde{O}(m^{1+o(1)} + n\epsilon^{-(c+1)})$ on expanders simply by running the algorithm on $H$ instead of $G$.

| Table 1: Effective Resistance Sketch | | | | Table 2: Effective Resistance Estimation | | |
|---|---|---|---|---|---|---|
| Citation | $\mathcal{T}_p$ | $\mathcal{T}_q$ | $s$ | Citation | Runtime | $S$ Restrictions |
| FMM | $n^\omega$ | $1$ | $n^2$ | FMM | $n^\omega + |S|$ | None |
| Solver | $nm$ | $1$ | $n^2$ | Solver | $nm + |S|$ | None |
| [5] | $m\epsilon^{-2}$ | $\epsilon^{-2}$ | $n\epsilon^{-2}$ | [5] | $n^2\epsilon^{-2}$ | $S = V \times V$ |
| [28] | $n^2\epsilon^{-1}$ | $1$ | $n\epsilon^{-1}$ | [28] | $n^2\epsilon^{-1}$ | $S = V \times V$ |
| [29] | $m + n\epsilon^{-2}$ | $1$ | $n\epsilon^{-1}$ | [19] | $m + (n + |S|)\epsilon^{-2}$ | None |
| **This Paper** | $m\epsilon^{-1}$ | $1$ | $n\epsilon^{-1}$ | [37] | $m + |S|\,\epsilon^{-2}$ | None |
| | | | | [30] | $m\epsilon^{-1.5}$ | $S = E$ |
| | | | | [29] | $m + n\epsilon^{-2} + |S|$ | None |
| | | | | **This Paper** | $m\epsilon^{-1} + |S|$ | None |

Summary of prior work on Effective Resistance Sketch (Table 1) and Effective Resistance Estimation (Table 2) algorithms on $n$-node, $m$-edge expanders. All time and space complexities are reported up to to $\widetilde{O}(\cdot)$. The methods of This Paper and [29] apply only to expanders; however, the remaining works apply to general graphs. As discussed, when $m^{1+o(1)} + n\epsilon^{-(c+1)} = o(m\epsilon^{-c})$, any runtime dependence on $m\epsilon^{-c}$ in the table can be improved to a dependence on $m^{o(1)+1} + n\epsilon^{-(c+1)}$.

**Lower Bounds** For the effective resistance estimation problem, [33] showed that any combinatorial algorithm (i.e., an algorithm that uses only combinatorial operations in the sense of Theorem 1) which solves the effective resistance estimation problem for $S = V \times V$ in $\widetilde{O}(n^2\epsilon^{-1/13+\delta})$ for some $\delta > 0$, would imply a combinatorial subcubic deterministic algorithm which detects a triangle in an $n$-node undirected unweighted graph. We improve on their result, as follows.

**Theorem 4.** *Given a combinatorial algorithm which solves the effective resistance estimation problem for $S = V \times V$ on graphs with $\widetilde{\Omega}(1)$-expansion in $\widetilde{O}(n^2\epsilon^{-1/2+\delta})$ time for $\delta > 0$, we can produce a randomized combinatorial algorithm which solves the triangle detection problem on an $n$-node graph in $\widetilde{O}(n^{3-2\delta})$ time whp.*

Theorem 4 implies an $\widetilde{\Omega}(n^2\epsilon^{-1/2})$ randomized conditional lower bound for the problem of estimating effective resistances of all pairs of nodes in an undirected unweighted expander graph, while [33] shows only an $\Omega(n^2\epsilon^{-1/13})$ lower bound. As we show in Section 4, by a simple reduction we can extend any $\widetilde{\Omega}(n^2\epsilon^{-c})$ lower bound for the all-pairs effective resistance problem to a $\widetilde{\Omega}(m\epsilon^{-c})$ lower bound for the all-edges effective resistance problem. Consequently, our result also yields a $\widetilde{\Omega}(m\epsilon^{-1/2})$ randomized lower bound for the problem of estimating effective resistances of all edges in an undirected expander graph.

In addition to conditional lower bounds for effective resistance estimation, we also improve on existing conditional lower bounds for the problem of estimating spectral sums that we define below.

**Definition 5** (Spectral Sum). *For $f : \mathbb{R}^+ \to \mathbb{R}^+$ and $\mathbf{A} \in \mathbb{R}^{n \times n}$ with singular values $\sigma_1(\mathbf{A}) \leqslant \sigma_2(\mathbf{A}) \leqslant \cdots \leqslant \sigma_n(\mathbf{A})$, we define the spectral sum $\mathcal{S}_f : \mathbb{R}^{n \times n} \to \mathbb{R}^+$ as $\mathcal{S}_f(\mathbf{A}) := \sum_{i=1}^n f(\sigma_i(\mathbf{A}))$.*

[33] showed that for several spectral sums $\mathcal{S}_f$, any combinatorial algorithm that outputs $Y \approx_\epsilon \mathcal{S}_f(\mathbf{A})$ in $(n^\gamma \epsilon^{-c})$ time on an $n \times n$ PSD matrix would imply an $O(n^{\gamma+\alpha c})$ time combinatorial algorithm which solves the triangle detection problem, where the scaling $\alpha$ varies depending on the specific $\mathcal{S}_f$ (see Table 3). We build on their results to show improved randomized conditional lower bounds for several spectral sum estimation problems, as presented in Theorem 5 below.

**Theorem 5.** *Given a combinatorial algorithm which on input $\mathbf{B} \in \mathbb{R}^{n \times n}$ outputs a spectral sum estimate $Y \approx_\epsilon \mathcal{S}_f(\mathbf{B})$ in $O(n^\gamma \epsilon^{-c})$ time with $\gamma \geqslant 2$ for the spectral sums in Table 3, we can produce a randomized combinatorial algorithm that can detect a triangle in an n-node graph whp. in $\widetilde{O}(n^{\gamma+\alpha c})$ time, where $\alpha$ is a scaling that depends on properties of the function $f$ (see Table 3 for values of $\alpha$ for several spectral sums.)*

| Spectral Sum | [33] | | This Paper | |
| --- | --- | --- | --- | --- |
| | TD Runtime | Lower Bound | TD Runtime | Lower Bound |
| Schatten 3-norm | $n^{\gamma+4c}$ | $n^2\epsilon^{-1/4}$ | $n^{\gamma+5c/2}$ | $n^2\epsilon^{-2/5}$ |
| Schatten $p$-norm, $p \neq 1, 2$, | $n^{\gamma+13c}$ | $n^2\epsilon^{-1/13}$ | $n^{\gamma+10c}$ | $n^2\epsilon^{-1/10}$ |
| SVD Entropy | $n^{\gamma+6c}$ | $n^2\epsilon^{-1/6}$ | $n^{\gamma+5c}$ | $n^2\epsilon^{-1/5}$ |
| Log Determinant | $n^{\gamma+6c}$ | $n^2\epsilon^{-1/6}$ | $n^{\gamma+5c}$ | $n^2\epsilon^{-1/5}$ |
| Trace of Exponential | $n^{\gamma+13c}$ | $n^2\epsilon^{-1/13}$ | $n^{\gamma+10c}$ | $n^2\epsilon^{-1/10}$ |

Table 3: Runtimes for the triangle detection (TD) problem in an $n$-node graph using algorithms that produce $(1 \pm \epsilon)$ multiplicative approximations to various spectral sums in $O(n^\gamma \epsilon^{-c})$ time. The second columns contain the best achievable runtimes for $\gamma = 2$ that do not use FMM, barring a breakthrough in subcubic triangle detection. All runtimes are reported up to $\tilde{O}(\cdot)$.

## 2 Preliminaries

**General notation.** We use $\mathbf{A}_{i,j}$ to denote the $(i,j)$-th entry of $\mathbf{A}$. For $\mathbf{A} \in \mathbb{R}^{n \times n}$ we use $\lambda(\mathbf{A})$ for its spectrum, $\lambda_i(\mathbf{A})$ and $\sigma_i(\mathbf{A})$ for its $i$-th smallest eigenvalue and singular value respectively, and $\rho(\mathbf{A}) := |\lambda_n(\mathbf{A})|$ for its spectral radius. $\| \cdot \|_p$ denotes the $\ell_p$-norm. When $\mathbf{A}$ is PSD, $\lambda_{\min}(\mathbf{A})$ denotes its smallest nonzero eigenvalue and $\kappa(\mathbf{A}) := \lambda_n(\mathbf{A})/\lambda_{\min}(\mathbf{A})$ denotes its pseudo-condition number. We use $\langle \cdot, \cdot \rangle$ for the Euclidean inner product, $\mathbb{1}$ for the all ones vector, and $e_i$ for the $i$-th standard basis vector. We define $\vec{\delta}_{i,j} := e_i - e_j$ and $[k] := \{1, ..., k\}$. We use $x \approx_\epsilon y$ as shorthand for $(1-\epsilon)y \leqslant x \leqslant (1+\epsilon)y$. For $v \in \mathbb{R}^n$, we use $v[i:j]$ for the sub-vector from index $i$ to $j$. We use $v \perp w$ to indicate that $v, w \in \mathbb{R}^n$ are orthogonal (i.e., $v^\top w = 0$).

**Graphs.** We use $G = (V, E, w)$ to denote a weighted undirected graph on $V$ with edges $E$ and edge weights $w \in \mathbb{R}^E_{\geqslant 0}$ (or $G = (V, E)$ if unweighted). We use $\mathbf{A}_G$ to denote its (weighted) adjacency matrix $(\mathbf{A}_G)_{u,v} = w_{u,v}$ for $u, v \in V \times V$ and $\mathbf{D}_G$ to denote its diagonal (weighted) degree matrix $(\mathbf{D}_G)_u = \sum_{\{u,v\} \in E} w_{u,v}$ for $u \in V$ (treated as $w_{u,v} = 1$ if $G$ is unweighted). We define $\mathbf{L}_G := \mathbf{D}_G - \mathbf{A}_G$ as its graph Laplacian. $d_{\max}(G)$ and $d_{\min}(G)$ refer to the max and min diagonal entry in $\mathbf{D}_G$. We may drop the argument or subscript $G$ if it is clear from context. The effective resistance between nodes $i, j \in V$ is denoted $r_G(i,j) = \vec{\delta}_{i,j}^\top \mathbf{L}_G^\dagger \vec{\delta}_{i,j}$. We assume all input graphs are connected, as effective resistances can be computed separately on connected components. We use $\mathbf{B}_G$ to denote the $E \times V$ edge-incidence matrix of $G$, where $(\mathbf{B}_G)_{e,u} = 1$ and $(\mathbf{B}_G)_{e,v} = -1$ for all $e \in E$, and $e = \{u, v\}$, with $\mathbf{B}_{Ge,l} = 0$ for all $l \neq u$ and $l \neq v$.

**Symmetric Diagonally Dominant (SDD) Matrices** A matrix $\mathbf{M} \in \mathbb{R}^{n \times n}$ is SDD if it can be decomposed as $\mathbf{M} = \mathbf{D_M} - \mathbf{A_M}$, where the $\mathbf{D_M}$ is a diagonal matrix with non-negative entries and $\mathbf{A}_M$ is a matrix with zeros on the diagonal such that $d_{i,i} > \sum_{j=1}^n |a_{i,j}|$. We define the normalization of $\mathbf{M}$ as $\mathbf{N}_M := \mathbf{D}_M^{-1/2} \mathbf{M} \mathbf{D}_M^{-1/2}$. Throughout this paper, we assume, without loss of generality that $\mathbf{D}_M$ has strictly positive entries on the diagonal (otherwise, we can simply remove an entire row and column of zeros). We use $d_{\max}(\mathbf{M})$ and $d_{\min}(\mathbf{M})$ to denote the max and min entry in the diagonal of $\mathbf{D}_M$ respectively. We may drop the argument or subscript $\mathbf{M}$ if it is clear from context.

**Spectral Graph Theory.** Our results leverage well-known spectral graph theory results. In particular, our algorithm complexities are parameterized by the normalized pseudo condition number of a graph (or SDD matrix).

**Definition 6** (Normalized (pseudo-)condition number). *We define the* normalized (pseudo-)condition number *of an SDD matrix* $\mathbf{M} \in \mathbb{R}^n$ *as* $\bar{\kappa}(\mathbf{M}) := \lambda_n(\mathbf{N}_M)/\lambda_{\min}(\mathbf{N}_M)$.

To connect the normalized condition number to expander graphs, we can apply Cheeger's inequality, which guarantees that if $G$ has $\alpha$-expansion for some $\alpha = \tilde{\Omega}(1)$, then $\bar{\kappa}(\mathbf{L}_G) = \lambda_n(\mathbf{N}_{L_G})/\lambda_2(\mathbf{N}_{L_G}) \leqslant 4/\alpha^2 = \tilde{O}(1)$.

**Theorem 6** (Cheeger's Inequality [38]). *Let* $G = (V, E, w)$ *be an undirected graph. Then,* $\frac{1}{2}\lambda_2(\mathbf{N}_L) \leqslant \phi(G) \leqslant \sqrt{2\lambda_2(\mathbf{N}_L)}$.

In addition, we leverage the fact that effective resistances can be expressed in several equivalent expressions. In particular, it is known that:

$$r_G(i,j) = \vec{\delta}_{i,j}^\top \mathbf{L}_G^\dagger \vec{\delta}_{i,j} = \vec{\delta}_{i,j} D^{-1/2} \mathbf{N}_G^\dagger D^{-1/2} \vec{\delta}_{i,j} = (\mathbf{W}_G^{1/2} \mathbf{B}_G \mathbf{L}_G^\dagger \vec{\delta}_{i,j})^\top (\mathbf{W}_G^{1/2} \mathbf{B}_G \mathbf{L}_G^\dagger \vec{\delta}_{i,j}),$$

where $\mathbf{W}_G \in \mathbb{R}^{E \times E}$ is the diagonal matrix of weights in $G$ [5].

**Runtimes and Space Complexities.** In our algorithmic results and analysis, when clear from context, we use $\widetilde{O}(\cdot)$ (resp., $\widetilde{\Omega}(\cdot)$) notation to hide polylogarithmic factors (resp., inverse polylogarithmic factors) in the number of vertices, the number of edges, the size of the matrix, the number of nonzero entries in a matrix, the maximum and minimum diagonal element of a matrix, the maximum and minimum weighted degree of a graph, $\epsilon$, the condition number, and the normalized psuedo-condition number of a matrix. We say event $E$ occurs with high probability in $t$ if $\mathbb{P}[E] \geqslant 1 - t^{-c}$, where $c > 0$ can be controlled by appropriately configuring the algorithm parameters. We may simply say that an event occurs "with high probability" or "whp." if it occurs with high probability in the size of a matrix or number of nodes in a graph.

## 3 Algorithmic Results

In this section, we present our main algorithmic results. Section 3.1 outlines our approach to effective resistance sketches and estimation; we defer discussion of our approach to SDD and PSD extensions to the supplementary material. Section 3.2, presents our original results on effective resistance sketching and estimation and generalizations to SDD matrices. Section 3.3 extends our techniques to yield an interesting data structure for estimating quadratic forms of PSD matrices.

### 3.1 Our Approach

**Approach in prior work: Johnson Lindenstrauss sketches.** Our starting inspiration is a classic result of [5], which obtains an $(\widetilde{O}(m\epsilon^{-2}), \widetilde{O}(\epsilon^{-2}), \widetilde{O}(n\epsilon^{-2}))$-effective resistance sketch by using the Johnson Lindenstrauss Lemma (JL) [39] and its algorithmic instantiations [40].

**Lemma 1** (Johnson-Lindenstrauss Lemma [40]). *Given fixed vectors $v_1, ..., v_n \in \mathbb{R}^d$ and $\epsilon \in (0,1)$, let $\mathbf{J}$ be an independently sampled random matrix in $\{\pm 1/\sqrt{k}\}^{k \times d}$. For $k = \widetilde{O}(\log(n)\epsilon^{-2})$, whp. in $n$, $\|\mathbf{J} v_i\|_2 \approx_\epsilon \|v_i\|_2$ for all $i \in [n]$.*

[5] observe that $r_G(i,j) = (\mathbf{W}_G^{1/2} \mathbf{B}_G \mathbf{L}_G^\dagger \vec{\delta}_{i,j})^\top (\mathbf{W}_G^{1/2} \mathbf{B}_G \mathbf{L}_G^\dagger \vec{\delta}_{i,j})$. Consequently, w.h.p in $n$, $\|\mathbf{J} \mathbf{W}_G^{1/2} \mathbf{B}_G \mathbf{L}_G^\dagger \vec{\delta}_{i,j}\|_2 \approx_\epsilon r_G(i,j)$. With SDD linear system solvers, $\mathbf{J} \mathbf{W}_G^{1/2} \mathbf{B}_G \mathbf{L}_G^\dagger$ can be approximated in $\widetilde{O}(m\epsilon^{-1})$ time, from which $\|\mathbf{J} \mathbf{W}_G^{1/2} \mathbf{B}_G \mathbf{L}_G^\dagger \vec{\delta}_{i,j}\|_2$ can be queried in $\widetilde{O}(\epsilon^{-2})$ time.

**Our approach: asymmetric CountSketch in $\ell_1$.** Towards improving upon JL sketches for effective resistance estimation, our key tool is to use other sketching algorithms. In particular we use that there are algoirthms that achieve better than $\widetilde{O}(\epsilon^{-2})$-sketch dimension for vectors with small $\ell_1$ norm with comparable guarantees, e.g., CountSketch. CountSketch is a classic memory-efficient algorithm for estimating the number of occurences of various datapoints in a data stream [41] and efficiently computing inner products [42]. Given $v \in \mathbb{R}^n$ and integer parameters $s, t > 0$, CountSketch transforms $v$ to a vector $\mathbf{S} v \in \mathbb{R}^{3ts \times n}$, where $\mathbf{S} \in \mathbb{R}^{3ts \times n}$ is a $3t$-column-sparse 0/1 matrix. Lemma 2 is a special case of Theorem 4 from [42], which provides accuracy guarantees for inner product estimation using CountSketch.

**Lemma 2** (Special Case of [42], Theorem 4). *Let vectors $v, w \in \mathbb{R}^n$. Let $\mathbf{S}$ be a random CountSketch matrix. Let $x_i = \langle (\mathbf{S}v)[(i-1)s + 1 : i \cdot s], (\mathbf{S}w)[(i-1)s + 1 : i \cdot s] \rangle$ for $i \in [3t]$, and let $X$ denote the median of $\{x_i\}$. For $s = O\left(\min\left(\frac{\|v\|_1 \|w\|_1}{\epsilon |\langle v,w \rangle|}, \frac{\|v\|_2^2 \|w\|_2^2}{\epsilon^2 |\langle v,w \rangle|^2}\right)\right)$, and $t = \log(n^c)$, $|X - \langle v,w \rangle| \leqslant \epsilon |\langle v,w \rangle|$ with probability at least $\Omega(1 - n^{-c})$.*

To improve the guarantee in Lemma 2 to hold whp. *for all $v, w \in S$* rather than for each fixed pair, one can simply choose $t = \log(n^c|S|)$ and apply a union bound. Consequently, if we knew that $\|\mathbf{W}_G^{1/2} \mathbf{B}_G \mathbf{L}_G^\dagger \vec{\delta}_{i,j}\|_1^2 / r_G(i,j) = \widetilde{O}(1)$, then building a CountSketch with $s = \widetilde{O}(\epsilon^{-1})$ would

yield a $\widetilde{O}(n\epsilon^{-1})$-size sketch, improving over the $\widetilde{O}(n\epsilon^{-2})$ sketch obtained using the $\ell_2$ JL sketch in [5]. Unfortunately, it is unclear if and when such a bound holds, and so, it is unclear how the $\ell_1$ CountSketches could be useful in this setting. This leads to the main insight that fuels our algorithms. Rather than seeking a symmetric factorization of $r_G(i,j)$ as a quadratic form $v^\top v$ and applying a sketching procedure to $v$, we instead work with an *asymmetric factorization*. In particular, we observe

$$r_G(i,j) = \langle \mathbf{D}_{\mathbf{L}_G}^{-1}\vec{\delta}_{i,j}, \mathbf{D}_{\mathbf{L}}^{1/2}(\mathbf{N}_{\mathbf{L}_G})^\dagger \mathbf{D}_{\mathbf{L}_G}^{-1/2}\vec{\delta}_{i,j}\rangle. \tag{1}$$

At first glance, it may seem unclear why (1) is helpful. However, we show that indeed, for expanders

$$\left\|\mathbf{D}_{\mathbf{L}}^{-1}\vec{\delta}_{i,j}\right\|_1 \left\|\mathbf{D}_{\mathbf{L}}^{1/2}(\mathbf{N}_{\mathbf{L}})^\dagger \mathbf{D}_{\mathbf{L}}^{-1/2}\vec{\delta}_{i,j}\right\|_1 / r_G(i,j) = \widetilde{O}(1). \tag{2}$$

Our main result essentially follows from (2). Using SDD linear system solvers, we can efficiently approximate $\mathbf{SD}_{\mathbf{L}}^{1/2}(\mathbf{N}_{\mathbf{L}})^\dagger \mathbf{D}_{\mathbf{L}}^{-1/2} \in \mathbb{R}^{\widetilde{O}(\epsilon^{-1})\times n}$ in $\widetilde{O}(m\epsilon^{-1})$ time, yielding our $\mathcal{T}_{\mathrm{q}}$ of $\widetilde{O}(m\epsilon^{-1})$ and $s$ of $\widetilde{O}(n\epsilon^{-1})$ (see discussion in supplementary material). Moreover, $\mathbf{SD}_{\mathbf{L}}^{-1}$ is $\widetilde{O}(1)$-sparse, due to the structure of Count-Sketch. So, using our (approximate) access to $\mathbf{SD}_{\mathbf{L}}^{1/2}(\mathbf{N}_{\mathbf{L}})^\dagger \mathbf{D}_{\mathbf{L}}^{-1/2}$, for any query $i,j \in V$, we can efficiently approximate (1) using the recovery procedure of Lemma 2 in $\widetilde{O}(1)$ time.

### 3.2 Our Results

We use the approach in Section 3.1 to develop algorithms to compute spectral sketch data structures for SDD matrices $\mathbf{M}$ with $\widetilde{O}(1)$ normalized condition number, as defined in Definitions 6 and 7. So, as discussed in Section 2, this implies that our spectral sketch algorithms will automatically apply to normalized Laplacians of expanders and enable us to compute effective resistances.

**Definition 7** (Spectral Sketch Data Structure). *We say an algorithm produces an $(\mathcal{T}_{\mathrm{s}}, \mathcal{T}_{\mathrm{q}}, s)$-spectral sketch data structure for a PSD matrix $\mathbf{A} \in \mathbb{R}^{n\times n}$ if given $\mathbf{A} \in \mathbb{R}^{n\times n}$ and $\epsilon \in (0,1)$, the algorithm creates a binary string of length $O(s(A,\epsilon))$ in time $O(\mathcal{T}_{\mathrm{s}}(A,\epsilon)$, from which, for any supported query $b \in \mathbb{R}^n$, w.h.p it outputs $q_A(b) \approx_\epsilon b^\top \mathbf{A}b$ in time $O(\mathcal{T}_{\mathrm{q}}(A,\epsilon)(\mathrm{nnz}(b))^2)$.*

Our spectral sketches of SDD matrices $\mathbf{M}$ will only support $\mathbf{D}_M$-numerically sparse query vectors.

**Definition 8** ($\mathbf{D}$-numerically sparse). *For a diagonal matrix $\mathbf{D}$, the $\mathbf{D}$-numerical sparsity of $x \in \mathbb{R}^n$ is $\mathrm{ns}_{\mathbf{D}}(x) := \left\|\mathbf{D}^{-1}x\right\|_1 \|x\|_1 / \left\|\mathbf{D}^{-1/2}x\right\|_2^2$. We say $x$ is $(c,\mathbf{D})$-numerically sparse if $\mathrm{ns}_{\mathbf{D}}(x) \leqslant c$.*

Definition 8 is restrictive; however, several natural classes of vectors satisfy the requirements. For example, $\mathbb{1}_i$ are $(1,\mathbf{D})$-numerically sparse and $\vec{\delta}_{i,j}$ is $(2,\mathbf{D}_M)$-numerically sparse for any $i,j \in [n]$ and invertible diagonal matrix $\mathbf{D} \in \mathbb{R}^{n\times n}$ (see supplementary material for additional examples.)

The following asymmetric rearrangement of quadratic forms is crucial to our analysis.

**Lemma 3.** *Let $\mathbf{M} \in \mathbb{R}^{n\times n}$ be SDD and $x \in \mathbb{R}^n$ be orthogonal to $\ker(\mathbf{M})$. Then,*

$$x^\top \mathbf{M}^\dagger x = \frac{1}{2}\left\langle \mathbf{D}_M^{-1}x, \mathbf{D}_M^{1/2}(\mathbf{N}_M/2)^\dagger \mathbf{D}_M^{-1/2}x\right\rangle = \left\langle \mathbf{D}_M^{-1/2}x, \mathbf{N}_M^\dagger \mathbf{D}_M^{-1/2}x\right\rangle \geqslant \frac{1}{2}\left\|\mathbf{D}_M^{-1/2}x\right\|_2^2.$$

*Proof.* For notational convenience, let $\mathbf{N}_{\ell M} = \mathbf{N}_M/2$. Let $v = \mathbf{M}^\dagger x = (\mathbf{D}_M - \mathbf{A}_M)^\dagger x$. Note that $\mathbf{D}_M^{-1/2}x \perp \ker(\mathbf{N}_M)$, and $2\mathbf{D}_M^{1/2}\mathbf{N}_{\ell M}\mathbf{D}_M^{1/2} = \mathbf{M}$. Consequently, $v = \frac{1}{2}\mathbf{D}_M^{-1/2}\mathbf{N}_{\ell M}^\dagger \mathbf{D}_M^{-1/2}x$, and hence $x^\top \mathbf{M}^\dagger x = \frac{1}{2}\langle \mathbf{D}_M^{-1}x, \mathbf{D}_M^{1/2}\mathbf{N}_{\ell M}^\dagger \mathbf{D}_M^{-1/2}x\rangle$. The second equality now follows immediately by rearranging terms. To obtain the inequality, note that, because $\mathbf{M}$ is SDD and $\mathbf{D}$ is invertible, $\mathbf{N}_M$ is PSD. Furthermore, $2\mathbf{I} - \mathbf{N}_M = \mathbf{I} - \mathbf{D}_M^{-1/2}\mathbf{A}_M\mathbf{D}_M^{-1/2}$, which is also PSD, as $\lambda(\mathbf{A}_M) \subset [-d_{\max}, d_{\max}]$. So, $0 \leqslant \lambda(\mathbf{N}_M) \leqslant 2$. So, $\lambda_{\min}(\mathbf{N}_M) \geqslant 1/2$ and the lemma follows. $\qquad\square$

Lemma 4 bounds $\left\|\mathbf{D}_M^{1/2}(\mathbf{N}_M/2)^\dagger \mathbf{D}_M^{-1/2}x\right\|_1$. The proof uses the power series expansion of $(\mathbf{N}_M/2)^\dagger$.

**Lemma 4.** *Let $\mathbf{M} \in \mathbb{R}^{n\times n}$ be an SDD matrix and $x \in \mathbb{R}^n$ be a unit vector orthogonal to $\ker(\mathbf{M})$. Then*

$$\left\|\mathbf{D}_M^{1/2}(\mathbf{N}_M/2)^\dagger \mathbf{D}_M^{-1/2}x\right\|_1 \leqslant \max\left(1, 2\overline{\kappa}(\mathbf{M})\log\left(\sqrt{nd_{\max}}2\overline{\kappa}(\mathbf{M})/\sqrt{d_{\min}}\right)\right)\|x\|_1 + 1.$$

Combining our Lemmas 3 and 4 with the guarantees of Lemma 2 and prior work on SDD linear system solvers (see supplementary material), we obtain the following theorem.

**Theorem 7.** *For any SDD matrix* $\mathbf{M} \in \mathbb{R}^{n \times n}$, *there is an algorithm to construct an* $(\widetilde{O}(\overline{\kappa}(\mathbf{M}) \operatorname{nnz}(\mathbf{M})\epsilon^{-1}), \widetilde{O}(1), \widetilde{O}(\overline{\kappa}(\mathbf{M}) n\epsilon^{-1}))$*-spectral sketch data structure of* $\mathbf{M}^{\dagger}$ *supported over queries* $S$, *where* $S$ *is any set of* $(\widetilde{O}(1), \mathbf{D}_M)$*-numerically sparse vectors orthogonal to* $\ker(\mathbf{M})$.

Because the $\vec{\delta}_{i,j}$ queries appearing in effective resistance computations are 2-sparse and $(2, \mathbf{D}_M)$-numerically sparse for all SDD matrices $\mathbf{M}$, taking $\mathbf{M} = \mathbf{L}_G$ in Theorem 7 immediately implies Theorem 2 and Theorem 3 (see supplementary material for detailed discussion and pseudocode.)

### 3.3 Extensions to PSD Matrices

Our approach of approximating quadratic forms via asymmetric inner products also yields a query-efficient sketching procedure for approximating quadratic forms of well-conditioned PSD matrices.

**Theorem 8.** *There is an algorithm to construct an* $(\widetilde{O}(\kappa(\mathbf{A})\operatorname{nnz}(\mathbf{A})\epsilon^{-2}), \widetilde{O}(1), \widetilde{O}(\kappa(\mathbf{A})n\epsilon^{-2}))$*-spectral sketch data structure of* $\mathbf{A}$ *supported over* $S$, *where* $S$ *is any set of vectors orthogonal to* $\ker(\mathbf{A})$ *and* $\mathbf{A}$ *is PSD.*

In comparison, JL [39] gives an $(\widetilde{O}(n^{\omega}), \widetilde{O}(\epsilon^{-2}), \widetilde{O}(n\epsilon^{-2}))$-spectral sketch data structure using efficient square root algorithms [43]. JL achieves better compression than Theorem 8, while Theorem 8 achieves faster query time. When the matrix is well conditioned and error tolerance is sufficiently high, Theorem 8 may achieve better construction time and query time than JL while maintaining comparable compression.

## 4 Lower Bounds

In this section, we present our main hardness results. Section 4.1 outlines our approach. In Section 4.2, we present our lower bounds for the problem of estimating effective resistances for all pairs of nodes (case where $S = V \times V$), which we call the "all pairs effective resistance estimation problem." Section 4.3 shows our techniques also yield lower bounds for other spectral sum estimation problems.

### 4.1 Our Approach

**Approaches of Previous Work** The approach of [33] begins with the fact that $G$ has a triangle if and only if $\operatorname{tr}(\mathbf{A}_G^3)/6 \geqslant 1$. They use the fact that various spectral sums $\mathcal{S}_f$ of the of the SDD matrix $\mathbf{I} - \delta\mathbf{A}_G$ (for $\delta$ sufficiently small) can be expressed as a power series $\mathcal{S}_f(\mathbf{I} - \delta\mathbf{A}_G) = \sum_{k=0}^{\infty} c_k \delta^k \operatorname{tr}(\mathbf{A}_G^k)$. The first two terms of this series can be computed directly. So given $Y \approx_{\epsilon} \mathcal{S}_f(\mathbf{I} - \delta\mathbf{A}_G)$, one can estimate $\operatorname{tr}(\mathbf{A}_G^3)$, where the estimation error is controlled by the magnitude of the first two terms of the series and the tail error due to truncating at the third term. By bounding this estimation error, [33] show that, for appropriate choices of $\delta$, $Y \approx_{\epsilon} \mathcal{S}_f(\mathbf{I} - \delta\mathbf{A}_G)$ yields an additive 1/2 approximation to $\operatorname{tr}(\mathbf{A}_G^3)$, which is sufficient for triangle detection. They also reduce the problem of estimating the spectral sum $\operatorname{tr}(\mathbf{B}^{-1})$ for an SDD matrix $\mathbf{B}$ to the all pairs effective resistance estimation problem.

**Our Approach** We use three key techniques to better bound the estimation errors incurred in the power-series-inspired approach of [33]. This yields faster reductions and better lower bounds for effective resistance estimation. Rather than obtaining effective resistance lower bounds by reducing the problem of computing $\operatorname{tr}(\mathbf{A}_G^3)/6$ to computing the *trace* of an SDD matrix as in [33], we use a reduction that closely resembles the structure of effective resistances. For $\alpha > 0$ sufficiently small,

$$\left(\mathbf{I} - \frac{\alpha}{n}\mathbf{A}_G\right)^{-1} = \sum_{k=0}^{\infty} \frac{\alpha^k}{n^k}\mathbf{A}_G^k. \tag{3}$$

Since $\mathbf{A}_G$ is known, given access to $\vec{\delta}_{i,j}^{\top}(\mathbf{I} - \frac{\alpha}{n}\mathbf{A}_G)^{-1}\vec{\delta}_{i,j}$, we can estimate the entries of $\mathbf{A}_G^2$, where the estimation error is controlled by $\alpha$ and the tail error of truncating (3) at the third term. By bounding this estimation error, for appropriate choice of $\alpha$, we can obtain additive 1/2 approximations to all entries of $\mathbf{A}_G^2$, which is sufficient to identify all paths of length 2. We can then detect a

triangle by simply scanning for an edge $\{u, v\}$ such that $u$ and $v$ are connected by a path of length 2. Estimating the entries of $\mathbf{A}_G^2$ leads to lower estimation error than estimating $\operatorname{tr}\left(\mathbf{A}_G^3\right)$ as in [33]). Second, we use a standard randomized reduction that reduces the triangle detection problem to the triangle detection problem restricted to tripartite graphs. The reduction relies on the fact that a randomly sampled tripartition of the original graph preserves triangles with constant probability. To detect a triangle in a tripartite graph $G = (V_1 \sqcup V_2 \sqcup V_3, E)$, we construct a graph $H$ by removing all edges $E_{1,2} := \{\{u, v\} \in E : u \in V_1, v \in V_2\}$ between $V_1$ and $V_2$. $G$ has a triangle if and only if there is an edge $\{u, v\} \in E_{1,2}$ and a path of length 2 between $u$ and $v$ in $H$. Crucially, we can show that the third term $\bar{\mathbf{A}}_H^3$ does not contribute to the tail error when estimating the $\{u, v\}$-th entry of $\left(\mathbf{A}_H^2\right)$ using (3). Third, to lower the spectral norm of $\mathbf{A}_H$ (and consequently better bound the convergence of the power series (3)), we introduce the symmetric random signing of $\mathbf{A}_H$ below.

**Definition 9** (Symmetric Random Signing). *Given a symmetric matrix $\mathbf{A} \in \mathbb{R}^{n \times n}$, its symmetric random signing $\bar{\mathbf{A}}$ is the random matrix with $\bar{\mathbf{A}}_{i,j} := \xi_{i,j} \mathbf{A}_{i,j}$, where $\xi_{i,j}$ are independent Rademacher random variables that satisfy $\xi_{i,j} = \xi_{j,i}$.*

We show that this random signing preserves the non-zeroness of entries of $\mathbf{A}_H^2$ with constant probability, allowing us to detect whether $G$ has a triangle even if we replace $\mathbf{A}_G$ in (3) with $\bar{\mathbf{A}}_H$ instead. This is beneficial, as matrix Chernoff guarantees $\left\|\bar{\mathbf{A}}_H\right\|_2 = \widetilde{O}(\sqrt{n})$ whp. whereas $\|\mathbf{A}_H\|_2$ may be as large as $n$. So the tail error of truncating the power series is smaller. To compute entries of $\bar{\mathbf{A}}_H^2$ efficiently using effective resistance estimates on expanders, we first show that we can use all pairs effective resistance estimates on expanders to estimate $\vec{\delta}_{i,j}^T \mathbf{M}^{-1} \vec{\delta}_{i,j}$ for all $i, j \in [n]$, where $\mathbf{M} = (\mathbf{I} - \mathbf{Q})$ is an SDD matrix with $\rho(\mathbf{Q}) \leqslant 1/3$. Then, by choosing $\mathbf{M} = \mathbf{I} - \frac{\alpha}{n} \bar{\mathbf{A}}_H$ as in (3) for an appropriate constant $\alpha$, we can estimate $\bar{\mathbf{A}}_H^2$ from estimates of $\vec{\delta}_{i,j}^T \mathbf{M}^{-1} \vec{\delta}_{i,j}$. This yields our lower bound on the all pairs effective resistance estimation problem.

Additionally, we show that random signing also preserves the non-zeroness of $\operatorname{tr}\left(\mathbf{A}_G^3\right)$ with constant probability, and leverage this to obtain improved randomized conditional lower bounds for various spectral sum estimation problems. We closely follow the trace estimation approach of [33], and again use the smaller spectral radius of $\bar{\mathbf{A}}_G$ to improve bounds on the power series truncation error.

## 4.2 Improved Lower Bounds for Effective Resistance Estimation

In this section we provide a series of reductions which yield our main result on lower bounds for the all pairs effective resistance estimation problem for all pairs of nodes (case where $S = V \times V$).

**Definition 10.** *In the SDD effective resistance estimation problem, given an SDD matrix $\mathbf{M}$ such that $\mathbf{D_M} = \mathbf{I}$, $\mathbf{A_M} = \mathbf{Q}$, with $\rho(\mathbf{Q}) \leqslant 1/3$ and $\epsilon \in (0, 1)$, we must output $\widetilde{r} \in \mathbb{R}^{n^2}$ such that $\widetilde{r}_{a,b} \approx_\epsilon \vec{\delta}_{a,b}^\top \mathbf{M}^{-1} \vec{\delta}_{a,b} \ \forall a, b \in [n]$. We call $\vec{\delta}_{a,b}^\top \mathbf{M}^{-1} \vec{\delta}_{a,b}$ the SDD effective resistance of $(a, b)$ in $\mathbf{M}$.*

For brevity, we use $\widetilde{r}(\mathbf{M})$ to refer to the solution of the SDD effective resistance problem on input $\mathbf{M}$. Our first step is to show that an algorithm for the all pairs effective resistance estimation problem on expanders implies an algorithm for the SDD effective resistance estimation problem.

**Lemma 5.** *Given an algorithm to solve the all pairs effective resistance estimation problem on graphs with $\widetilde{\Omega}(1)$-expansion in $\widetilde{O}(n^2 \epsilon^{-c})$ time for some $c > 0$, we can produce an algorithm to solve the SDD effective resistance estimation problem in $\widetilde{O}(n^2 \epsilon^{-c})$ time.*

To prove Lemma 5, we first prove the lemma for the case where $\mathbf{Q}$ is entrywise non-negative by constructing an expander $G$ with $n + 1$ vertices such that $r_G(a, b) = \vec{\delta}_{a,b}^\top \mathbf{M}^{-1} \vec{\delta}_{a,b}$ for all $a, b \in [n]$. We extend the reduction to arbitrary $\mathbf{Q}$ by constructing $\mathbf{Q}'$ of size $2n$ so that $\mathbf{Q}'$ is entrywise non-negative and $\widetilde{r}(\mathbf{I} - \mathbf{Q})$ is a simple linear transform of $\widetilde{r}(\mathbf{I} - \mathbf{Q}')$.

We turn our attention to reducing the triangle detection problem to the SDD effective resistance problem. As discussed, a key aspect of our approach is to work with the random signing $\bar{\mathbf{A}}_G$ of $\mathbf{A}_G$. Lemma 6 shows that to determine whether $(\mathbf{A}_G^2)_{i,j} > 0$ with constant probability, it suffices to determine whether $(\bar{\mathbf{A}}_G^2)_{i,j} > 0$. Matrix Chernoff ensures whp. $\rho(\bar{\mathbf{A}}_G) = \widetilde{O}(\sqrt{n})$, while $\rho(\mathbf{A}_G)$ could be as large as $n$ [44]. So, estimating entries of $\bar{\mathbf{A}}_G$ leads to lower power series tail error.

**Lemma 6.** *For $i \neq j$, if $(\mathbf{A}_G^2)_{i,j} = 0$, then $(\bar{\mathbf{A}}_G^2)_{i,j} = 0$; if $(\mathbf{A}_G^2)_{i,j} > 0$, $\mathbb{P}\left[|(\bar{\mathbf{A}}_G^2)_{i,j}| > 1\right] \geqslant 1/2$.*

The idea of the proof is that if $\{a,b\}, \{b,c\}$ exist in $G$, either $\xi_{a,c} = 1$ or $\xi_{a,c} = -1$ results in $(\bar{\mathbf{A}}_G^2)_{a,c} > 0$. Finally, we use the power series approach in Section 4.1 to obtain Theorem 9.

**Theorem 9.** *Given an algorithm which solves the SDD effective resistance estimation problem in $\widetilde{O}(n^2\epsilon^{-c})$ time, we can produce a randomized algorithm that solves the triangle detection problem in $\widetilde{O}(n^{2(1+c)})$ time whp.*

Theorem 9 and Lemma 5 with $c = 1/2 - \delta$ immediately imply our main result Theorem 4.

Additionally, we extend the lower bound of Theorem 4 to the all edges effective resistance estimation by the following reduction.

**Lemma 7.** *Given an algorithm to solve the all edges effective resistance estimation problem (i.e., Definition 1 where $S = E$) in $\widetilde{O}(m\epsilon^{-c})$ time, we can produce an algorithm to solve the all pairs effective resistance estimation problem in $\widetilde{O}(n^2\epsilon^{-c})$ time for some $c > 0$.*

The rough idea behind the reduction is to add a complete graph of edges of sufficiently small weight that would not change the effective resistances much. Lemma 7 combined with Theorem 4 then imply a $\widetilde{\Omega}(m\epsilon^{-1/2})$ randomized lower bound for the all edges effective resistance estimation problem on graphs with graphs with $\widetilde{\Omega}(1)$-expansion.

### 4.3 Improved Lower Bounds for Spectral Sum Estimation

Finally, we discuss our improved lower bounds for various spectral sum estimation problems. Analogous to Lemma 6, in the following lemma we show that to determine whether a graph has a triangle (i.e., $\operatorname{tr}\left(\mathbf{A}_G^3\right) > 0$) with constant probability, it suffices to determine whether $\operatorname{tr}\left(\bar{\mathbf{A}}_G^3\right) > 0$.

**Lemma 8.** *If $\operatorname{tr}\left(\mathbf{A}_G^3\right) = 0$, then $\operatorname{tr}\left(\bar{\mathbf{A}}_G^3\right) = 0$, and if $\operatorname{tr}\left(\mathbf{A}_G^3\right) > 0$ then $\mathbb{P}\left[\left|\operatorname{tr}\left(\bar{\mathbf{A}}_G^3\right)\right| > 0\right] \geqslant 1/4$.*

The central idea of the proof is that if a triangle $\{a,b\}, \{b,c\}, \{c,a\}$ exists in $G$, then amongst the 4 possible configurations of the Rademacher random signing variables $\xi_{a,b}$ and $\xi_{b,c}$, at least one configuration must result in $\left|\operatorname{tr}\left(\bar{\mathbf{A}}_G^3\right)\right| > 0$. By following the proof of Theorem 15 from [33], and replacing their use of $\mathbf{A}_G$ with a symmetric random signing $\bar{\mathbf{A}}_G$, we obtain an improved randomized version of their result by leveraging the smaller spectral radius of $\bar{\mathbf{A}}_G$. Theorem 5 follows by applying this result to the functions $f$ that define the corresponding spectral sums (see supplementary material).

## 5   Conclusion

In this paper we provided improved upper and lower bounds on the problem of estimating and sketching effective resistances on expanders. On the algorithmic side we show how sketches tailored to $\ell_1$ when carefully applied to asymmetric formulations of the quadratic form of the Laplacian pseudoinverse gave our results. On the lower bound side, we provided an alternative to the trace estimation approach of [33] for showing lower bounds and coupled it with techniques of randomly signing edges of the graph to obtain our results. Further, we showed that these techniques had broader implications for addressing algorithmic challenges in numerical linear algebra.

Beyond the natural open problem of improving both our upper and lower bounds towards bringing them together, there are interesting open problems in broadening the applicability of both our upper and lower bounds. For example, obtaining an $\widetilde{O}(m\epsilon^{-1})$ time algorithm for estimating the effective resistance of all edges in a general (non-expander) graph and extending our $\widetilde{\Omega}(n^2\epsilon^{-1/2})$ lower bounds to deterministic algorithms remain interesting open problems. We hope that the results of this paper provide useful tools for addressing each.

## Acknowledgments and Disclosure of Funding

We thank Hongyue Li for helpful discussions and work on this project at various stages. Aaron Sidford was supported in part by a Microsoft Research Faculty Fellowship, NSF CAREER Award CCF-1844855, NSF Grant CCF-1955039, a PayPal research award, and a Sloan Research Fellowship. Ishani Karmarkar was supported in part by an NSF CAREER Award CCF-1844855, NSF Grant

CCF-1955039, a PayPal research grant, and a Stanford Institute for Computational and Mathematical Engineering (ICME) Fellowship. Rajat Vadiraj Dwaraknath was supported by a Stanford ICME Fellowship.

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
