# 6 Supplementary Material

In this supplementary material section, we include additional discussion and proofs of the results from the main body. In Section 6.1, we discuss additional related work on effective resistance estimation and fine-grained complexity analysis more broadly. In Section 6.2, we discuss additional details and provide omitted proofs pertaining to our algorithmic results in Section 3. In Section 6.3, we discuss additional details and provide omitted proofs pertaining to our hardness results in Section 4.

## 6.1 Additional Related Work

Here we supplement the discussion in Section 1 by briefly discussing additional work related to the effective resistance estimation problem and providing a more detailed comparison of our results to [29].

**Dynamic effective resistance estimation.** Effective resistance estimation and sketching are part of a broader family of previously studied problems involving graph compression and effective resistance estimation. For example, there is a related line of work on dynamically maintaining effective ressistance estimates in dynamic graphs, e.g., [19, 31], which in turn is related to problems of dynamically maintaining electric flows in graphs, e.g., [45, 46]. Whether our techniques have ramifications for these related problems is an interesting question for future work.

**Fine-grained complexity analysis.** Our effective resistance estimation lower bounds fall under a broader topic of fine grained complexity analysis, i.e., the problem of characterizing the optimal complexity of problems which are known to have polynomial time solutions. Here, we provide references to a few examples. [34] showed subcubic equivalences between the problem of triangle detection, Boolean matrix multiplication, and several other graphical problems. As discussed, [33] utilize the results of [34] to obtain conditional lower bounds on several spectral sum approximation problems– many of which we also study in this paper. Similarly, [47–49] provided several conditional complexity lower bounds for linear algebraic problems, conditional on the use of particular computational models. [50] and [51] also provide fine grained lower bounds for the fault replacement paths problem, problems on graph centrality measures, and complementary problems. Making connections between our techniques for our effective resistance estimation lower bounds and these prior works in fine-grained complexity analysis is an interesting open problem.

**Comparison with the approach of [29].** Effective reisistance sketching and estimation for expanders was previously studied in [29]. [29] produces an $(\widetilde{O}(m + n\epsilon^{-2}), \widetilde{O}(1), \widetilde{O}(n\epsilon^{-1}))$ effective resistance sketch for graphs with $\widetilde{\Omega}(1)$-expansion. Our work provides a different, independently obtained runtime for effective resistance estimation by producing an $(\widetilde{O}(m\epsilon^{-1}), \widetilde{O}(1), \widetilde{O}(n\epsilon^{-1}))$ effective resistance sketch for graphs with $\widetilde{\Omega}(1)$ expansion. Additionally, our work can be applied to produce an $(\widetilde{O}(m^{1+o(1)} + n\epsilon^{-2}), \widetilde{O}(1), \widetilde{O}(n\epsilon^{-1}))$ effective resistance sketch, by running on a sparse graphical sketch, such as that guaranteed by [30] (see Section 1.1). Consequently, our results match those of [29] up to an $m^{o(1)}$ factor in all regimes, and improve for sufficiently high accuracy on sparse graphs.

Given a graph $G$ with $\widetilde{\Omega}(1)$-expansion, [29] proposes an algorithm which is motivated by the idea of storing $\widetilde{O}(1)$-sparse approximations to the columns of $\mathbf{L}_G^\dagger$, which would clearly be sufficient for querying effective resistances of $G$ in $\widetilde{O}(1)$ time. However, it is unclear whether the columns of $\mathbf{L}_G^\dagger$ have small $\ell_1$ norm, and consequently, it is unclear how to obtain these sparse approximations. Consequently, their algorithm instead estimates the following vector $\sigma_u$ for each $u \in V$,

$$\sigma_u = \frac{1}{2} \sum_{t=0}^{\infty} \left( \left( \frac{1}{2}\mathbf{I} + \frac{1}{2}\mathbf{A}_G\mathbf{D}_G^{-1} \right)^t e_u - \pi \right),$$

where $\pi = \frac{\mathbf{D}_{\mathbf{L}_G}\mathbb{1}}{\mathbb{1}^\top \mathbf{D}_{\mathbf{L}_G}\mathbb{1}}$. They show that $\sigma_u$ is closely related to $\mathbf{D}_G\mathbf{L}_G^\dagger e_u - \pi$, and consequently access to $\sigma_u$ is sufficient for estimating effective resistances. Additionally, they use structural properties of expanders to show that each $\sigma_u$ must have small $\ell_1$ norm and that it can be computed efficiently by running lazy random walks on $G$ (i.e., the random walk which, at each step follows the natural

random walk on $G$ with probability 1/2 and stays idle with probability 1/2). These key properties of $\sigma_u$ enable their result.

Our approach is similar to that of [29] in that we also reformulate the effective resistance between two vertices as an inner product between two vectors whose $\ell_1$ norm we can bound; however, the vectors we consider are not the $\sigma_u$ vectors considered in [29]. Instead, we rewrite $r_G(i,j)$ as the inner product of $\mathbf{D}_{\mathbf{L}_G}^{-1}\vec{\delta}_{i,j}$ with $\mathbf{D}_{\mathbf{L}_G}^{1/2}(\mathbf{N}_{\mathbf{L}_G}/2)^\dagger\mathbf{D}_{\mathbf{L}_G}^{-1/2}\vec{\delta}_{i,j}$. Similar to [29], we then use similar underlying structural properties of expanders to argue that the $\ell_1$ norms of these vectors is not too large. However, instead of using random walks to estimate these vectors, we use sketching techniques (specifically, CountSketch) and Laplacian system solvers to estimate them, an idea which is inspired by the work of [5]. The differences in the specific effective resistance vectors we estimate and the different technique of estimating them is what leads to the difference in runtime between [29] and our own work. [3] [29] also provide an extension of their effective resistance sketch techniques to well-conditioned SDD matrices, which we also do in our generalized Theorem 7; our lower bounds on the SDD effective resistance estimation problem (see Section 4.2) therefore also apply to the work of [29].

## 6.2 Additional Discussion of Algorithmic Results

In Section 6.2.1, we discuss related background on CountSketch and SDD linear system solvers, which are crucial to our proofs of our upper bound results. In Section 6.2.2 we expand on our discussion of $\mathbf{D}_{\mathbf{M}}$-numerical sparsity from Definition 8 and provide our motivation for studying such queries. In Section 6.2.3 we present omitted proofs from Section 3.2 and Section 3.3.

### 6.2.1 Relevant Technical Background

**CountSketch.** Below we prove that Lemma 2 follows from Theorem 4 in [42].

*Proof of Lemma 2.* Applying Theorem 4 from [42] with $t = 2$,

$$\mathbb{V}\left[|X - \langle v, w \rangle|\right] \leqslant \min\left(3\frac{\|v\|_1^2\|w\|_1^2}{s^2}, 2\frac{\|v\|_2^2\|w\|_2^2}{s}\right).$$

Applying Markov's inequality for the second moment,

$$\mathbb{P}\left[|X - \langle v, w \rangle| > 2\sqrt{3}\min\left(\frac{\|v\|_1\|w\|_1}{s}, \frac{\|v\|_2\|w\|_2}{\sqrt{s}}\right)\right] \leqslant \frac{1}{4}.$$

Consequetly, setting $s = O\left(\min\left(\frac{\|v\|_1\|w\|_1}{\epsilon\langle v,w\rangle}, \frac{\|v\|_2^2\|w\|_2^2}{\epsilon^2\langle v,w\rangle^2}\right)\right)$ suffices for $\mathbb{P}\left[|X - \langle v, w \rangle| \leqslant \epsilon\langle v, w \rangle\right] \geqslant 3/4$. To improve the failure probability to $O(n^{-c})$, it suffices to repeat the sketch $O\left(\log(n^c)\right)$ times and output the median. □

**SDD Linear System Solvers.** In order to compute our effective resistance sketches efficiently, we apply a CountSketch matrix $\mathbf{S}$ to $\mathbf{M}^\dagger$, where $\mathbf{M}$ is an SDD matrix. To do this efficiently, we leverage the following theorem.

**Theorem 10** (SDD Linear System Solver). *Let $\mathbf{M} \in \mathbb{R}^{n \times n}$ be SDD and consider any $\beta > 0$. There exists a randomized algorithm which, with high probability in $n$, processes a graph in time $\widetilde{O}(\mathrm{nnz}\,(\mathbf{M}))$ to create access to a linear operator $\mathbf{Q}_\beta \in \mathbb{R}^{n \times n}$ such that $\mathbf{Q}_\beta$ can be applied to any $b \in \mathbb{R}^n$ with $b \perp \ker(\mathbf{M})$ in time $\widetilde{O}(m)$ and $\|\mathbf{Q}_\beta b - \mathbf{M}^\dagger b\|_{\mathbf{M}} \leqslant \beta\|\mathbf{M}^\dagger b\|_{\mathbf{M}}$.*

Many SDD linear system solvers can be viewed as the type of an operator $\mathbf{Q}_\beta$ required in Theorem 10. For a particular example in which this is apparent, consider the operator corresponding to the iterative solver proposed in [52] or the solver from [53]. There is a long line of research on nearly linear time SDD and Laplacian system solvers, beginning with the work of [54] and leading to current state-of-the-art randomized algorithm of [55].

---

[3] More precisely, when writing the dependence on $\bar{\kappa}(G) := \bar{\kappa}(\mathbf{L}_G)$, [29] has an $\widetilde{O}(m + n\epsilon^{-2}\bar{\kappa}(G)^3 + |S|)$ runtime for the effective resistance estimation problem. This paper instead presents a runtime of $\widetilde{O}(m\epsilon^{-1}\bar{\kappa}(G) + |S|)$ for the effective resistance estimation problem.

### 6.2.2 Examples of $\mathbf{D}_M$-numerical Sparsity

As we discussed, our spectral sketch data structures in Section 3.2 allow for a restricted set of queries to the pseudoinverse, in particular those that are $\mathbf{D}_M$-numerically sparse as defined in Definition 8. Here, we elaborate on types of queries that are $\mathbf{D}_M$-numerically sparse.

- $\vec{\delta}_{i,j}$ is $(2, \mathbf{D})$-numerically sparse for any invertible $\mathbf{D} \in \mathbb{R}^{n \times n}$. As we are interested primarily in effective resistance estimation in this paper, this provides the primary motivation for studying this class of vectors.

- Standard basis vectors are $(1, \mathbf{D})$-numerically sparse for any invertible $\mathbf{D} \in \mathbb{R}^{n \times n}$.

- When $\mathbf{D}$ is the identity matrix, Definition 8 reduces to the standard definition of numerical sparsity [56].

- More generally, if $x \in \mathbb{R}^n$ is $\gamma$-numerically sparse, then it is $\left( \gamma \frac{\max_i d_{i,i}}{\min_i d_{i,i}}, \mathbf{D} \right)$-numerically sparse for any diagonal $\mathbf{D} \in \mathbb{R}^{n \times n}$. Note that if $\mathbf{D}$ is approximately a multiple of the identity, then the $\mathbf{D}_M$-numerical sparsity is approximately equal to the numerical sparsity, up to constants.

The approaches discussed in Section 3.2 apply to all of these examples.

### 6.2.3 Omitted Proofs form Section 3

In this section, we provide proofs of omitted results from Section 6.2.3. For notational convenience, in this section we define $\mathbf{A}_{\ell M} := \frac{1}{2}\mathbf{I} + \frac{1}{2}\mathbf{D}_M^{-1/2}\mathbf{A}_M \mathbf{D}_M^{-1/2}$. Note that $\mathbf{N}_M = 2(\mathbf{I} - \mathbf{A}_{\ell M})$.

First, let us prove Lemma 4. We first prove the following lemma regarding the power series expansion of $\mathbf{N}_{\ell M} := (\mathbf{N}_M/2)^\dagger$.

**Lemma 9.** *Let $\mathbf{M} \in \mathbb{R}^{n \times n}$ be SDD. For any $x \perp \ker(M)$,*

$$(\mathbf{N}_M/2)^\dagger \mathbf{D}_M^{-1/2} x = \sum_{k=0}^{\infty} (\mathbf{A}_{\ell M})^k \mathbf{D}_M^{-1/2} x,$$

*and for $m \geqslant 1$,*

$$\left\| (\mathbf{N}_M/2)^\dagger \mathbf{D}_M^{-1/2} x - \sum_{k=0}^{m} (\mathbf{A}_{\ell M})^k \mathbf{D}_M^{-1/2} x \right\|_1 \leqslant \frac{\sqrt{n} 2\bar{\kappa}(M)}{\sqrt{d_{\min}}} \exp\left( -\frac{m+1}{2\bar{\kappa}(M)} \right) \|x\|_2.$$

*Proof.* Let $r$ denote the rank of $M$. Let $q_1, ..., q_n$ denote orthonormal eigenvectors of $\mathbf{A}_{\ell M}$ associated with $\lambda_1(\mathbf{A}_{\ell M}), ..., \lambda_n(\mathbf{A}_{\ell M})$ respectively, $\mathbf{Q}$ denote the orthogonal matrix whose $i$-th column is $q_i$, and $\widetilde{\Lambda}$ denote the diagonal matrix of the $\lambda_i(\mathbf{A}_{\ell M})$'s.

An orthogonal eigendecomposition of $(\mathbf{N}_M/2)^\dagger$ is given by $(\mathbf{N}_M/2)^\dagger = \mathbf{Q}\Lambda\mathbf{Q}^\top$, where $\Lambda$ is the diagonal matrix whose entries are given by

$$\Lambda_{i,i} = \begin{cases} 0, & i = r+1, ..., n \\ \frac{1}{1-\lambda_i(\mathbf{A}_{\ell M})}, & i = 1, ..., r \end{cases}.$$

So, for any $t \in [r]$, $(\mathbf{N}_M/2)^\dagger q_t = \frac{1}{1-\lambda_t(\mathbf{A}_{\ell M})} q_t$, where $\lambda_t(\mathbf{A}_{\ell M}) \in (0,1)$; and consequently, $0 \leqslant \lambda_t(\mathbf{A}_{\ell M}) < 1$ is in the radius of convergence for the power series of $\frac{1}{1-x}$, and hence

$$\sum_{k=0}^{\infty} (\mathbf{A}_{\ell M})^k = \sum_{k=0}^{\infty} \mathbf{Q}\widetilde{\Lambda}^k \mathbf{Q}^\top q_t = q_t \sum_{k=0}^{\infty} \lambda_t^k = \frac{1}{1 - \lambda_t(\mathbf{A}_{\ell M})} q_t.$$

Now, $x \perp \ker(M)$ implies $\mathbf{D}_M^{-1/2} x \perp \ker(\mathbf{N}_M)$; and consequently, $\mathbf{D}_M^{-1/2} x$ can be expressed as a linear combination of $q_1, ..., q_r$. The first statement in the lemma now follows by linearity.

For the second statement, note that $\lambda_{\min}(\mathbf{N}_{\ell M}) = \lambda_n(\mathbf{N}_{\ell M})/\bar{\kappa}(M) \geqslant 1/(2\bar{\kappa}(M))$, so $\lambda_r(\mathbf{A}_{\ell M}) \leqslant 1 - 1/(2\bar{\kappa}(M))$. Since $\mathbf{D}_M^{-1/2} x \perp q_{r+1}, ..., q_n$,

$$\left\| (\mathbf{A}_{\ell M})^k \mathbf{D}_M^{-1/2} x \right\|_2 \leqslant \left( 1 - \frac{1}{2\bar{\kappa}(M)} \right)^k \frac{\|x\|_2}{\sqrt{d_{\min}}}.$$

Using the fact that $\|x\|_1 \leqslant \sqrt{n}\|x\|_2$ for all $x$, for $m \geqslant 1$, we have

$$\left\|(1/2\mathbf{N}_M)^\dagger \mathbf{D}_M^{-1/2}x - \sum_{k=0}^{m}(\mathbf{A}_{\ell M})^k \mathbf{D}_M^{-1/2}x\right\|_1 \leqslant \sqrt{n}\left\|(1/2\mathbf{N}_M)^\dagger \mathbf{D}_M^{-1/2}x - \sum_{k=0}^{m}(\mathbf{A}_{\ell M})^k \mathbf{D}_M^{-1/2}x\right\|_2$$

$$\leqslant \sqrt{n}\sum_{k=m+1}^{\infty}\left\|(\mathbf{A}_{\ell M})^k \mathbf{D}_M^{-1/2}x\right\|_2$$

$$\leqslant \frac{\sqrt{n}\,2\bar{\kappa}(M)}{\sqrt{d_{\min}}}\left(1 - \frac{1}{2\bar{\kappa}(M)}\right)^{m+1}\|x\|_2$$

$$\leqslant \frac{\sqrt{n}\,2\bar{\kappa}(M)}{\sqrt{d_{\min}}}\exp\left(-\frac{m+1}{2\bar{\kappa}(M)}\right)\|x\|_2.$$

$\square$

Using Lemma 9, the proof of Lemma 4 is now straightforward.

*Proof of Lemma 4.* Let $m \geqslant \max\left(1, 2\bar{\kappa}(M)\log\left(\frac{\sqrt{n d_{\max}}\,2\bar{\kappa}(M)}{\sqrt{d_{\min}}}\right)\right)$. By Lemma 9,

$$\left\|\mathbf{D}_M^{1/2}\mathbf{N}_{\ell M}^\dagger \mathbf{D}_M^{-1/2}x - \sum_{k=0}^{m}D^{1/2}\mathbf{A}_{\ell M}{}^k\mathbf{D}_M^{-1/2}x\right\|_1 = \left\|\mathbf{D}_M^{1/2}\left(\mathbf{N}_{\ell M}^\dagger \mathbf{D}_M^{-1/2}x - \sum_{k=0}^{m}\mathbf{A}_{\ell M}{}^k\mathbf{D}_M^{-1/2}x\right)\right\|_1$$

$$\leqslant \frac{\sqrt{n d_{\max}}\,2\bar{\kappa}(M)}{\sqrt{d_{\min}}}\exp\left(-\frac{m+1}{2\bar{\kappa}(M)}\right)\|x\|_2 \leqslant 1.$$

Using triangle inequality plus the observation that $\mathbf{D}_M^{1/2}\mathbf{A}_{\ell M}\mathbf{D}_M^{-1/2} = 1/2\mathbf{I} + 1/2\mathbf{A}_M\mathbf{D}_M^{-1}$ has each column normalized to have absolute column sum at most 1,

$$\left\|\mathbf{D}_M^{1/2}\mathbf{N}_{\ell M}^\dagger \mathbf{D}_M^{-1/2}x\right\|_1 \leqslant \left\|\sum_{k=0}^{m}\mathbf{D}_M^{1/2}\mathbf{A}_{\ell M}{}^k\mathbf{D}_M^{-1/2}x\right\|_1 + 1 \leqslant m\|x\|_1 + 1.$$

$\square$

Next, we will prove our result in Theorem 7. The corresponding algorithm pseudocode for constructing the spectral sketch data structure is given in Algorithm 1. The algorithm pseudocode for querying the spectral sketch data structure is given in Algorithm 2. The proof of Theorem 7 guarantees that it suffices to set $t = \widetilde{O}(1)$ and $s = \widetilde{O}(\bar{\kappa}(\mathbf{M})\epsilon^{-1})$ in Algorithm 1 and Algorithm 2. In the following, we use $\|x\|_{\mathbf{A}} := x^\top \mathbf{A}x$ to be the $\mathbf{A}$-seminorm for any PSD matrix $\mathbf{A}$.

*Proof of Theorem 7.* It suffices to assume, without loss of generality, that $S$ is a set of unit vectors, as at query time, for any vector $b$ we can compute $\|b\|_2$ in $O(\mathrm{nnz}(b))$ time and rescale. Set $\beta = \frac{\min(1, d_{\min}^3)\lambda_{\min}(\mathbf{M})\epsilon}{2\max(1, d_{\max}^2)\sqrt{n\max(1, d_{\max})}}$. By Theorem 10, in $\widetilde{O}(\mathrm{nnz}(\mathbf{M}))$, whp. we can obtain access to a linear operator $\mathbf{Q}_\beta$ such that $\mathbf{Q}_\beta$ can be applied to any $b \in S$ in time $\widetilde{O}(\mathrm{nnz}(\mathbf{M}))$ and $\|\mathbf{Q}_\beta b - \mathbf{M}^\dagger b\|_M \leqslant \beta\|\mathbf{M}^\dagger b\|_M = \beta\|b\|_{\mathbf{M}^\dagger}$. Then,

$$\lambda_{\min}(\mathbf{M})\|(\mathbf{Q}_\beta b - \mathbf{M}^\dagger)b\|_2^2 \leqslant \|(\mathbf{Q}_\beta b - \mathbf{M}^\dagger)b\|_M^2 \leqslant \beta^2\|b\|_{\mathbf{M}^\dagger}^2 \leqslant \frac{\beta^2}{\lambda_{\min}(\mathbf{M})}\|b\|_2^2.$$

So, $\|(\mathbf{Q}_\beta b - \mathbf{M}^\dagger)b\|_2 \leqslant \frac{\beta}{\lambda_{\min}(\mathbf{M})} \leqslant \epsilon$. Consequently, by triangle inequality, we have that

$$\|2\mathbf{D}_M\mathbf{Q}_\beta b\|_1 \leqslant \|2\mathbf{D}_M\mathbf{M}^\dagger b\|_1 + \|2\mathbf{D}_M\mathbf{Q}_\beta b - 2\mathbf{D}_M\mathbf{M}^\dagger b\|_1$$

where $2\mathbf{D}_M\mathbf{M}^\dagger b = \mathbf{D}_M^{1/2}(\mathbf{N}_M/2)^\dagger \mathbf{D}_M^{-1/2}b$ and $\|2\mathbf{D}_M\mathbf{Q}_\beta b - 2\mathbf{D}_M\mathbf{M}^\dagger b\|_1 \leqslant 2\sqrt{n d_{\max}}\|\mathbf{Q}_\beta b - \mathbf{M}^\dagger b\|_2 \leqslant \epsilon$. Consequently,

$$\|2\mathbf{D}_M\mathbf{Q}_\beta b\|_1 \leqslant \|\mathbf{D}_M^{1/2}(\mathbf{N}_M/2)^\dagger \mathbf{D}_M^{-1/2}b\|_1 + \epsilon.$$

Now, Lemma 4 guarantees that $\|2\mathbf{D}_M\mathbf{Q}_\beta b\|_1 \leqslant \widetilde{O}(\bar\kappa(\mathbf{M}) + \epsilon)\|b\|_1 = \widetilde{O}(\bar\kappa(\mathbf{M}))\|b\|_1$. Similarly,

$$
\left|\langle\mathbf{D}_M^{-1}b, \mathbf{D}_M^{1/2}(\mathbf{N}_M/2)^\dagger\mathbf{D}_M^{-1/2}b\rangle - \langle\mathbf{D}_M^{-1}b, 2\mathbf{D}_M\mathbf{Q}_\beta b\rangle\right| = \left|\langle\mathbf{D}_M^{-1}b, 2\mathbf{D}_M\mathbf{M}^\dagger b\rangle - \langle\mathbf{D}_M^{-1}b, 2\mathbf{D}_M\mathbf{Q}_\beta b\rangle\right|
$$

$$
\leqslant 2d_{\max}\left\|\mathbf{D}_M^{-1}b\right\|_2\left\|(\mathbf{Q}_\beta b - \mathbf{M}^\dagger)b\right\|_2
$$

$$
\leqslant 2\frac{d_{\max}}{d_{\min}}\frac{\beta}{\lambda_{\min}(\mathbf{M})} \leqslant \epsilon\left(\frac{1}{2d_{\min}^2}\right).
$$

Lemma 3 guarantees that $\langle\mathbf{D}_M^{-1}b, 2\mathbf{D}_M\mathbf{Q}_\beta b\rangle \geqslant O(1)\|\mathbf{D}_{\mathbf{M}}^{-1/2}b\|_2^2$. Consequently,

$$
\frac{\|\mathbf{D}_M^{-1}b\|_1\|2\mathbf{D}_M\mathbf{Q}_\beta b\|_1}{\langle\mathbf{D}_M^{-1/2}b, 2\mathbf{D}_M\mathbf{Q}_\beta b\rangle} = \widetilde{O}(\bar\kappa(\mathbf{M}))\frac{\|\mathbf{D}_M^{-1}b\|_1\|b\|_1}{\|\mathbf{D}_{\mathbf{M}}^{-1/2}b\|_2^2} = \widetilde{O}(\bar\kappa(\mathbf{M}))\mathrm{ns}_{\mathbf{D}_M}(b) = \widetilde{O}(\bar\kappa(\mathbf{M})).
$$

So, using $\tilde{O}(1)$ copies of a CountSketch matrix $\mathbf{S} \in \mathbb{R}^{\widetilde{O}(\bar\kappa(\mathbf{M})\epsilon^{-1})\times n}$, Lemma 2 guarantees that we can compute an $X$ such that whp.

$$
\left|X - \langle\mathbf{D}_M^{-1}b, 2\mathbf{D}_M\mathbf{Q}_\beta b\rangle\right| \leqslant \epsilon\langle\mathbf{D}_M^{-1}b, 2\mathbf{D}_M\mathbf{Q}_\beta b\rangle.
$$

Moreover, we showed above that

$$
\left|\langle\mathbf{D}_M^{-1}b, 2\mathbf{D}_M\mathbf{Q}_\beta b\rangle - \langle\mathbf{D}_M^{-1}b, \mathbf{D}_M^{1/2}(\mathbf{N}_M/2)^\dagger\mathbf{D}_M^{-1/2}b\rangle\right| \leqslant \epsilon\left(\frac{1}{2d_{\min}^2}\right)
$$

$$
\leqslant \epsilon\langle\mathbf{D}_M^{-1}b, \mathbf{D}_M^{1/2}(\mathbf{N}_M/2)^\dagger\mathbf{D}_M^{-1/2}b\rangle,
$$

where the last line uses the observation from Lemma 3, that $\langle\mathbf{D}_M^{-1}b, \mathbf{D}_M^{1/2}(\mathbf{N}_M/2)^\dagger\mathbf{D}_M^{-1/2}b\rangle \geqslant \frac{1}{2}\|\mathbf{D}_{\mathbf{M}}^{-1}b\|_2^2$. It now follows that

$$
\left|X - \langle\mathbf{D}_M^{-1}b, \mathbf{D}_M^{1/2}(\mathbf{N}_M/2)^\dagger\mathbf{D}_M^{-1/2}b\rangle\right| \leqslant \epsilon\langle\mathbf{D}_M^{-1}b, 2\mathbf{D}_M\mathbf{Q}_\beta b\rangle + \epsilon\langle\mathbf{D}_M^{-1}b, \mathbf{D}_M^{1/2}(\mathbf{N}_M/2)^\dagger\mathbf{D}_M^{-1/2}b\rangle
$$

$$
\leqslant 2\epsilon(1+\epsilon)\langle\mathbf{D}_M^{-1}b, \mathbf{D}_M^{1/2}(\mathbf{N}_M/2)^\dagger\mathbf{D}_M^{-1/2}b\rangle
$$

$$
\leqslant 4\epsilon\langle\mathbf{D}_M^{-1}b, \mathbf{D}_M^{1/2}(\mathbf{N}_M/2)^\dagger\mathbf{D}_M^{-1/2}b\rangle.
$$

Thus, by Lemma 3, $\frac{1}{2}X \approx_{4\epsilon} \langle\frac{1}{2}\mathbf{D}_M^{-1}b, \mathbf{D}_M^{1/2}(\mathbf{N}_M/2)^\dagger\mathbf{D}_M^{-1/2}b\rangle = \langle b, M^\dagger b\rangle$; hence, rescaling $\epsilon$ by a constant factor of 4 yields the approximation guarantee (without changing the size of $\mathbf{S}$ by more than constant factors).

Consequently, we see that by storing $\mathbf{S}\mathbf{D}_M\mathbf{Q}_\beta$, we can support queries $b \in S$. To justify the runtime guarantee, note that Theorem 10 shows we can compute $\mathbf{S}\mathbf{D}_M\mathbf{Q}_\beta$ in $\widetilde{O}(\mathrm{nnz}(\mathbf{M})\bar\kappa(\mathbf{M})\epsilon^{-1})$ time, by applying an approximate SDD system solver to each row in $\mathbf{S}$. To support queries, we need only store $\mathbf{S}\mathbf{D}_M\mathbf{Q}_\beta$ and $\mathbf{S}$, which requires only $\widetilde{O}(n\bar\kappa(\mathbf{M})\epsilon^{-1})$ bits.

Finally, we justify the query complexity. The key observation is that $\mathbf{S}$ is $\widetilde{O}(1)$-column sparse. Computing $X$ requires taking the median of $\widetilde{O}(1)$ quantities, each of which requires computing an inner product involving an $\widetilde{O}(\mathrm{nnz}(b))$-sparse vector $\mathbf{S}\mathbf{D}_M^{-1}b$. Using this fact, $X$ can be computed in $\widetilde{O}(\mathrm{nnz}(b)^2)$. □

We can now provide a proof of Theorem 2, which follows almost immediately from Theorem 7.

*Proof of Theorem 2.* Let $G = (V, E, w)$ be a graph with $\widetilde{\Omega}(1)$-expansion. As argued previously (see Section 3.2), $\mathbf{L}_G$ is SDD with $\widetilde{O}(1)$ normalized condition number, $\vec\delta_{i,j} \perp \ker\mathbf{L}_G$, and $\vec\delta_{i,j}$ is $(2, \mathbf{D}_{\mathbf{L}_G})$- numerically sparse. To see why Theorem 2 holds, simply observe that we can boost the whp. guarantee in Theorem 7, which holds for each fixed query, to hold whp. for all $\vec\delta_{i,j}$ for $i, j \in V$ by maintaining $O(\log(n))$ copies of the spectral sketch data structure guaranteed in Theorem 7 as our sketch, and then, at query time, taking the median of the results of the outputs from querying each of the $O(\log(n))$ copies of the sketch. □

---

**Algorithm 1:** SpectralSketchSDDInverse

---

**Input:** A matrix $\mathbf{M} \in \mathbb{R}^{n \times n}$, error tolerance $\epsilon \in (0,1)$, and integers $s, t > 0$.

**Output:** A CountSketch matrix $\mathbf{S} \in \mathbb{R}^{(3t)s \times n}$ and $\tilde{\mathbf{S}} \in \mathbb{R}^{(3t)s \times n}$, an approximation of
$2\mathbf{SD_M M}^\dagger$

1 Set $\beta$ as in the proof of Theorem 7;
2 Generate a random CountSketch matrix $\mathbf{S} \in \mathbb{R}^{(3t)s \times n}$ (see Theorem 4 of [42]) ;
3 Generate access to a linear operator $\mathbf{Q}_\beta$ such that for all $b \perp \ker(\mathbf{M})$,
   $\|\mathbf{Q}_\beta \mathbf{b} - \mathbf{M}^\dagger \mathbf{b}\|_M \leqslant \beta \|M^\dagger b\|_{\mathbf{M}}$ (See Theorem 10);
4 Compute $\tilde{\mathbf{S}} = 2\mathbf{SD_M Q}_\beta$;
5 **return** $(\mathbf{S}, \tilde{\mathbf{S}})$

---

---

**Algorithm 2:** QuerySketchSDDInverse

---

**Input:** $\mathbf{D_M}$, output of Algorithm 1, query vector $b$, and integers $s, t > 0$ as inputted to
   Algorithm 1.

**Output:** An approximation to $\langle b, \mathbf{M}^\dagger b \rangle$.

1 Set $\tilde{b} = b/\|b\|_2$;
2 **for** $i \in [3t]$ **do**
3 $\quad x_i = \frac{1}{2}\langle (\mathbf{SD}_M^{-1}\tilde{b})[(i-1)s + 1 : (i)s]i, (\tilde{\mathbf{S}}\tilde{b})[(i-1)s + 1 : (i)s]\rangle$ ;
4 **end**
5 **return** $\|b\|_2^2 \mathrm{median}\{x_i\}$.

---

Theorem 3 is now a direct corollary of Theorem 2. Finally, we provide the proof of Theorem 8.

*Proof of Theorem 8.* Let $x \perp \ker(\mathbf{A})$. Note that for any $\|x\|_2 \|\mathbf{A}x\|_2 = \kappa(\mathbf{A}^{1/2})x^\top \mathbf{A}x$. So, by the $\ell_2$ norm guarantees from Lemma 2, it suffices to build a CountSketch matrix $S$ with $s = \widetilde{O}\left(\kappa(\mathbf{A})\epsilon^{-2}\right)$ to guarantee that for any $x \perp \ker(\mathbf{A})$, $\langle \mathbf{S}x, \mathbf{S}\mathbf{A}x \rangle \approx_\epsilon \langle x, \mathbf{A}x \rangle$. The time to compute the sketch $\mathbf{S}\mathbf{A}$ is at most $\widetilde{O}(\kappa(\mathbf{A})\mathrm{nnz}(\mathbf{A})\epsilon^{-2})$. Due to the column-sparsity of $\mathbf{S}$, $\mathbf{S}x$ is $\widetilde{O}(\mathrm{nnz}(x))$ sparse, and consequently, $\langle \mathbf{S}x, \mathbf{S}\mathbf{A}x \rangle$ can be computed in $\widetilde{O}(\mathrm{nnz}(x)^2)$ time. □

### 6.3 Lower Bounds

Here we formalize the approach for lower bounds that was discussed in Section 4.

#### 6.3.1 Proofs for Lower Bounds on Effective Resistance Estimation

In this section, we present proofs that were omitted from Section 4.2. First, we formalize a standard randomized reduction from triangle detection in general graphs to triangle detection in tripartite graphs in Lemma 10 below.

**Lemma 10.** *Given an algorithm which can solve the triangle detection problem on an $n$-node tripartite undirected graph in $O(n^\gamma)$ time, we can produce a randomized algorithm which can solve the triangle detection problem on an arbitrary $n$-node undirected graph $G$ in $\widetilde{O}(n^\gamma)$ time whp.*

*Proof.* We first sample a tripartite subgraph $H$ of $G$ by assigning each vertex in $G$ to a random tripartition with equal probability 1/3 and deleting edges within each resulting tripartition. First, note that if $G$ has no triangles then $H$ also has no triangles since $H$ is a subgraph of $G$. Second, observe that if $G$ has a triangle $\{a, b, c\}$, it is also a triangle in $H$ if each vertex ends up in a different tripartition. This occurs with probability at least $(1/3)^3 = 1/27$ which is a constant. Therefore, solving the triangle detection problem on $H$ and returning the same output also successfully solves the triangle detection problem on $G$ with probability at least $1/27$. We can repeat this randomized procedure $\log(n^c)$ times to boost the success probability to at least $1 - n^{-c}$, which is whp. in $n$. This randomized algorithm runs in $\widetilde{O}(n^\gamma)$ time, completing the proof. □

Next, we establish some crucial properties of symmetric random signing that enables our results. First, we show in Lemma 11 below that the symmetric random signing of the adjacency matrix of a graph leads to a smaller spectral radius.

**Lemma 11.** *Let $G = (V, E)$ be an undirected unweighted graph on $n$ nodes. Let $\bar{\mathbf{A}}_G$ denote a symmetric random signing of $\mathbf{A}_G$. With high probability, $\rho(\bar{\mathbf{A}}) \leqslant \tilde{O}(\sqrt{n})$.*

*Proof.* Let $\sigma^2 := \left\| \sum_{\{u,v\} \in E} (\mathbf{E}_{u,v})^2 \right\|_2$ where $\mathbf{E}_{u,v}$ is the adjacency matrix of a graph with only a single edge between $u$ and $v$. Note that entries of $\mathbf{E}_{u,v}^2$ indicates paths of length two in this graph. Therefore, this is a diagonal matrix that satisfies $\left(\mathbf{E}_{u,v}^2\right)_{i,i} = 1$ if and only if $i \in \{u, v\}$. Consequently,

$$\sigma^2 = \left\| \sum_{\{u,v\} \in E} (\mathbf{E}_{u,v})^2 \right\|_2 = \|\mathbf{D}\|_2 \leqslant d_{\max} \leqslant n.$$

We can now write $\bar{\mathbf{A}} = \sum_{\{u,v\} \in E} \xi_{u,v} \mathbf{E}_{u,v}$ and apply the Matrix Rademacher concentration result (Theorem 1.2) from [44], to get that for any constant $c > 1$,

$$\mathbb{P}\left[ \lambda_n\left(\bar{\mathbf{A}}\right) \geqslant \sqrt{d_{\max}} \log(cn) \right] \leqslant n \exp\left( \frac{-cd_{\max} \log(n)}{2d_{\max}} \right) = n \frac{1}{n^c} = n^{-c+1}.$$

$\square$

Next, we prove Lemma 6 presented in Section 4.2 which states that the symmetric random signing preserves the non-zeroness of the entries of $\mathbf{A}_G^2$ with constant probability.

**Lemma 6.** *For $i \neq j$, if $(\mathbf{A}_G^2)_{i,j} = 0$, then $(\bar{\mathbf{A}}_G^2)_{i,j} = 0$; if $(\mathbf{A}_G^2)_{i,j} > 0$, $\mathbb{P}\left[|(\bar{\mathbf{A}}_G^2)_{i,j}| > 1\right] \geqslant 1/2$.*

*Proof.* Note that

$$\left(\bar{\mathbf{A}}^2\right)_{i,j} = \sum_{k=1}^{n} \bar{\mathbf{A}}_{i,k} \bar{\mathbf{A}}_{k,j}$$

If $\mathbf{A}_{i,j}^2 = 0$, then $G$ has no path of length exactly two between $i$ and $j$, so each term $\mathbf{A}_{i,k} \mathbf{A}_{k,j}$ in the summation above must be zero, and hence $\left(\bar{\mathbf{A}}^2\right)_{i,j} = 0$, completing the first part of the lemma.

Since $\left(\bar{\mathbf{A}}^2\right)_{i,j}$ is only supported on the integers, to prove the second statement, it suffices to show that when $\left(\mathbf{A}^2\right)_{i,j} > 0$, $\mathbb{P}\left[\left|\left(\bar{\mathbf{A}}^2\right)_{i,j}\right| = 0\right]$ is no larger than 1/2. To see this, note that if $\mathbf{A}_{i,j}^2 \neq 0$, then then $G$ has at least one path of length exactly 2 between $i$ and $j$. That is, there exists a $k' \in [n]$ such that $\mathbf{A}_{i,k'} \mathbf{A}_{k',j} = 1$. We can write

$$\left(\bar{\mathbf{A}}^2\right)_{i,j} = \sum_{k=1}^{n} \xi_{i,k} \xi_{k,j} \mathbf{A}_{i,k} \mathbf{A}_{k,j}$$

Because $i \neq j$, for every $k, \ell \in [n]$ each $\xi_{i,k}$ is always independent from any other $\xi_{\ell,j}$ term appearing in the sum. Moreover, if $\xi_{i,k}$ appears in the sum, then $\xi_{k,i}$ never appears in the sum. Therefore, for each $\{\xi_{i,k} \xi_{k,j}\}_{k \in [n]}$ are themselves independently chosen Rademacher random variables, and all terms in the summation are independent. Separating out the $k'$-th entry,

$$\left(\bar{\mathbf{A}}^2\right)_{i,j} = \xi_{i,k'} \xi_{k',j} + \sum_{k \neq k'}^{n} \xi_{i,k} \xi_{k,j} \mathbf{A}_{i,k} \mathbf{A}_{k,j} \overset{D}{=} \xi' + Z,$$

where $\xi'$ is a Rademacher random variable and $Z := \sum_{k \neq k'}^{n} \xi_{i,k} \xi_{k,j} \mathbf{A}_{i,k} \mathbf{A}_{k,j}$ is independent from $\xi'$. For any value of $Z$, $\mathbb{P}[\xi' = -Z] \leqslant \frac{1}{2}$. Therefore,

$$\mathbb{P}\left[\left(\bar{\mathbf{A}}^2\right)_{i,j} = 0\right] \leqslant \frac{1}{2},$$

completing the second part of the result.

$\square$

Now, we prove Lemma 5 in two steps. First, we show in Lemma 12 that given an algorithm to solve the all pairs effective resistance estimation problem on expanders, we can produce an algorithm to solve the SDD effective resistance estimation problem specifically on SDD matrices with non-positive offidiagonals. Then, we show in Lemma 14 that this algorithm can be used to produce an algorithm to solve the general SDD effective resistance estimation problem.

**Lemma 12.** *Given an algorithm that solves the all pairs effective resistance estimation problem on graphs with $\widetilde{\Omega}(1)$-expansion in $\widetilde{O}(n^2 \epsilon^{-c})$ time for some $c > 0$, we can produce an algorithm which takes as input an SDD matrix $\mathbf{M} = \mathbf{I} - \mathbf{Q}$ such that $\mathbf{Q}$ is entrywise non-negative and $\rho(\mathbf{Q}) \leqslant \frac{1}{3}$, and solves the SDD effective resistance estimation problem for $\mathbf{M}$ in $\widetilde{O}(n^2 \epsilon^{-c})$ time.*

*Proof.* Let $v := (\mathbf{I} - \mathbf{Q})\mathbb{1}$. Note that $v$ is entrywise non-negative. Consider the matrix

$$\mathbf{L} := \begin{pmatrix} \mathbf{I} & 0 \\ 0 & \|v\|_1 \end{pmatrix} - \begin{pmatrix} \mathbf{Q} & v \\ v^\top & 0 \end{pmatrix}.$$

and note that it is the Laplacian matrix. Since $\mathbf{M}$ is a principal submatrix of $\mathbf{L}$, we have by the eigenvalue interlacing theorem [57] that $\lambda_1(\mathbf{M}) \leqslant \lambda_2(\mathbf{L})$. But since $\rho(\mathbf{Q}) \leqslant 1/3$, we have that $\lambda_1(\mathbf{M}) \leqslant 2/3$ and consequently $\lambda_2(\mathbf{L}) \geqslant 2/3$. Therefore, $\mathbf{L}$ is the Laplacian of an $\widetilde{\Omega}(1)$-expander (see Cheeger's Inequality Theorem 6). We claim that for $i, j \in n$,

$$\vec{\delta}_{i,j}^\top \mathbf{L}^\dagger \vec{\delta}_{i,j} = \vec{\delta}_{i,j}^\top \mathbf{M}^{-1} \vec{\delta}_{i,j},$$

which is sufficient to prove the lemma. Note that for any $x \in \mathbb{R}^n$, $y \in \mathbb{R}$, and $\alpha \in \mathbb{R}$,

$$\mathbf{L} \begin{pmatrix} x + \alpha\mathbb{1} \\ y + \alpha \end{pmatrix} = \begin{pmatrix} (\mathbf{I} - \mathbf{Q})x + v^\top x \\ v^\top x - \|v\|_1 y \end{pmatrix}.$$

Let

$$\mathbf{L}^\dagger \vec{\delta}_{i,j} = \begin{pmatrix} z_x \\ z_y \end{pmatrix}.$$

Then, note that

$$\mathbf{L} \begin{pmatrix} z_x - z_y\mathbb{1} \\ 0 \end{pmatrix} = \begin{pmatrix} \vec{\delta}_{i,j} \\ 0 \end{pmatrix} \implies (\mathbf{I} - \mathbf{Q})(z_x - z_y\mathbb{1}) = \vec{\delta}_{i,j}.$$

Consequently,

$$\vec{\delta}_{i,j}^\top (\mathbf{I} - \mathbf{Q})^{-1} \vec{\delta}_{i,j} = \vec{\delta}_{i,j}^\top (z_x - z_y\mathbb{1}) = \vec{\delta}_{i,j}^\top z_x - z_y \vec{\delta}_{i,j}^\top \mathbb{1} = \vec{\delta}_{i,j}^\top z_x = \vec{\delta}_{i,j}^\top \mathbf{L}^\dagger \vec{\delta}_{i,j}.$$

$\square$

Now, to prove Lemma 14, we first show a useful property of block symmetric matrices in the helper lemma below.

**Lemma 13.** *Suppose $\mathbf{A} = \begin{pmatrix} \mathbf{X} & \mathbf{Y} \\ \mathbf{Y} & \mathbf{X} \end{pmatrix}$ where $\mathbf{X}, \mathbf{Y} \in \mathbb{R}^{n \times n}$. Then,*

$$\vec{\delta}_{i,j}^\top (\mathbf{X} - \mathbf{Y})\vec{\delta}_{i,j} = \frac{1}{2}\left[\vec{\delta}_{i,n+i}^\top \mathbf{A}\vec{\delta}_{i,n+i} + \vec{\delta}_{j,n+j}^\top \mathbf{A}\vec{\delta}_{j,n+j}\right] - \vec{\delta}_{i,n+j}^\top \mathbf{A}\vec{\delta}_{i,n+j} + \vec{\delta}_{i,j}^\top \mathbf{A}\vec{\delta}_{i,j}.$$

*Proof.* Note that

$$\vec{\delta}_{i,j}^\top (\mathbf{X} - \mathbf{Y})\vec{\delta}_{i,j} = (\mathbf{X}_{i,i} + \mathbf{X}_{j,j}) - 2\mathbf{X}_{i,j} + 2\mathbf{Y}_{i,j} - (\mathbf{Y}_{i,i} + \mathbf{Y}_{j,j}).$$

Meanwhile,

$$\frac{1}{2}\vec{\delta}_{i,n+i}^\top \mathbf{A}\vec{\delta}_{i,n+i} = (\mathbf{X}_{i,i} - \mathbf{Y}_{i,i}),$$

$$\frac{1}{2}\vec{\delta}_{j,n+j}^\top \mathbf{A}\vec{\delta}_{j,n+j} = (\mathbf{X}_{j,j} - \mathbf{Y}_{j,j}),$$

$$-\vec{\delta}_{i,n+j}^\top \mathbf{A}\vec{\delta}_{i,n+j} = -\mathbf{X}_{i,i} + 2\mathbf{Y}_{i,j} - \mathbf{X}_{j,j},$$

$$\vec{\delta}_{i,j}^\top \mathbf{A}\vec{\delta}_{i,j} = \mathbf{X}_{i,i} - 2\mathbf{X}_{i,j} + \mathbf{X}_{j,j},$$

and adding these four terms together concludes the proof. $\square$

**Lemma 14.** *Suppose we are given an algorithm which takes as input an SDD matrix $\mathbf{M} = \mathbf{I} - \mathbf{Q}$ such that $\mathbf{Q}$ is entrywise non-negative and $\rho(\mathbf{Q}) \leqslant \frac{1}{3}$, and solves the SDD effective resistance estimation problem for $\mathbf{M}$ in $\tilde{O}(n^2 \epsilon^{-c})$ time. Then, we can produce an algorithm which takes as input an SDD matrix $\mathbf{M}' = \mathbf{I} - \mathbf{Q}'$ and $\rho(\mathbf{Q}') \leqslant \frac{1}{3}$, and solves the SDD effective resistance estimation problem for $\mathbf{M}'$ in $\tilde{O}(n^2 \epsilon^{-c})$ time.*

*Proof.* We can decompose $\mathbf{Q}'$ as $\mathbf{Q}' = \mathbf{P} - \mathbf{N}$ where $\mathbf{P}$ is a matrix which contains only the positive offdiagonal entries of $\mathbf{Q}'$ and $-\mathbf{N}$ is a matrix which contains all the negative offdiagonal entries. Therefore both $\mathbf{P}$ and $\mathbf{N}$ themselves are entrywise non-negative. We define

$$\mathbf{Q} := \begin{pmatrix} \mathbf{P} & \mathbf{N} \\ \mathbf{N} & \mathbf{P} \end{pmatrix}.$$

Note that $\mathbf{Q}$ is also entrywise non-negative. We also have $\rho(\mathbf{Q}) \leqslant 1/3$. To see this, assume for the sake of contradiction that $\mathbf{Q}$ has an eigenvalue $\lambda > 1/3$. This means that there must exist some $x \in \mathbb{R}^{2n}$ such that $\mathbf{Q}x = \lambda x$. Let $x = [x_1; x_2]$ where $x_1, x_2 \in \mathbb{R}^n$. The eigenvalue equation then implies that $\mathbf{P}x_1 + \mathbf{N}x_2 = \lambda x_1$ and $\mathbf{N}x_1 + \mathbf{P}x_2 = \lambda x_2$. Subtracting these equations yields $(\mathbf{P} - \mathbf{N})(x_1 - x_2) = \lambda(x_1 - x_2)$ which means that $\lambda$ is also an eigenvalue of $\mathbf{Q}'$. This is a contradiction since $\rho(\mathbf{Q}') \leqslant 1/3$. Therefore, $\rho(\mathbf{Q}) \leqslant 1/3$.

Now, consider the following block decomposition of $\mathbf{Q}^k$

$$\mathbf{Q}^k = \begin{pmatrix} \mathbf{X} & \mathbf{Y} \\ \mathbf{Y} & \mathbf{X} \end{pmatrix}$$

where $\mathbf{X}, \mathbf{Y} \in \mathbb{R}^{n \times n}$. We will show by induction that $\mathbf{Q}'^k = (\mathbf{P} - \mathbf{N})^k = \mathbf{X} - \mathbf{Y}$. In the base case, when $k = 1$, this is trivially true. Now, assume that the claim holds for all $k \leqslant m$ for some $m$. Consider the following block decomposition of $\mathbf{Q}^m$

$$\mathbf{Q}^m = \begin{pmatrix} \mathbf{W} & \mathbf{Z} \\ \mathbf{Z} & \mathbf{W} \end{pmatrix}.$$

By the inductive hypothesis, we know that $\mathbf{Q}'^m = (\mathbf{P} - \mathbf{N})^m = \mathbf{W} - \mathbf{Z}$. Now, we have that

$$\mathbf{Q}^{m+1} = \begin{pmatrix} \mathbf{W} & \mathbf{Z} \\ \mathbf{Z} & \mathbf{W} \end{pmatrix} \begin{pmatrix} \mathbf{P} & \mathbf{N} \\ \mathbf{N} & \mathbf{P} \end{pmatrix} = \begin{pmatrix} \mathbf{WP} + \mathbf{ZN} & \mathbf{WN} + \mathbf{ZP} \\ \mathbf{WN} + \mathbf{ZP} & \mathbf{WP} + \mathbf{ZN} \end{pmatrix},$$

and also that

$$\mathbf{Q}'^{m+1} = (\mathbf{W} - \mathbf{Z})(\mathbf{P} - \mathbf{N}) = \mathbf{WP} - \mathbf{ZP} - \mathbf{WN} + \mathbf{ZN} = (\mathbf{WP} + \mathbf{ZN}) - (\mathbf{WN} + \mathbf{ZP}).$$

Hence, the claim follows by induction. By Lemma 13, it follows that

$$\vec{\delta}_{i,j}^\top \mathbf{Q}'^k \vec{\delta}_{i,j} = \frac{1}{2}\left[ \vec{\delta}_{i,n+i}^\top \mathbf{Q}^k \vec{\delta}_{i,n+i} + \vec{\delta}_{j,n+j}^\top \mathbf{Q}^k \vec{\delta}_{j,n+j} \right] - \vec{\delta}_{i,n+j}^\top \mathbf{Q}^k \vec{\delta}_{i,n+j} + \vec{\delta}_{i,j}^\top \mathbf{Q}^k \vec{\delta}_{i,j}. \quad (4)$$

Now we can use the power series expansion of $(\mathbf{I} - \mathbf{Q})^{-1}$ to say, for any $u, v \in [2n]$,

$$\vec{\delta}_{u,v}^\top (\mathbf{I} - \mathbf{Q})^{-1} \vec{\delta}_{u,v} = \sum_{k=0}^\infty \vec{\delta}_{u,v}^\top \mathbf{Q}^k \vec{\delta}_{u,v}.$$

Similarly, for any $i, j \in [n]$,

$$\vec{\delta}_{i,j}^\top (\mathbf{I} - \mathbf{Q}')^{-1} \vec{\delta}_{i,j} = \sum_{k=0}^\infty \vec{\delta}_{i,j}^\top \mathbf{Q}'^k \vec{\delta}_{i,j}.$$

So, by linearity and (4), it follows that

$$\vec{\delta}_{i,j}^\top (\mathbf{I} - \mathbf{Q}')^{-1} \vec{\delta}_{i,j} = \frac{1}{2}\left[ \vec{\delta}_{i,n+i}^\top (\mathbf{I} - \mathbf{Q})^{-1} \vec{\delta}_{i,n+i} + \vec{\delta}_{j,n+j}^\top (\mathbf{I} - \mathbf{Q})^{-1} \vec{\delta}_{j,n+j} \right]$$
$$- \vec{\delta}_{i,n+j}^\top (\mathbf{I} - \mathbf{Q})^{-1} \vec{\delta}_{i,n+j} + \vec{\delta}_{i,j}^\top (\mathbf{I} - \mathbf{Q})^{-1} \vec{\delta}_{i,j},$$

and this completes the proof. □

Lemma 5 follows directly from Lemma 12 and Lemma 14. Finally, we prove Theorem 9 below.

**Theorem 9.** *Given an algorithm which solves the SDD effective resistance estimation problem in $\widetilde{O}(n^2\epsilon^{-c})$ time, we can produce a randomized algorithm that solves the triangle detection problem in $\widetilde{O}(n^{2(1+c)})$ time whp.*

*Proof.* As $G$ is tripartite, let $V = V_1 \sqcup V_2 \sqcup V_3$ be the partition of $G$ such that no edge has both endpoints in $V_i$ for some $i \in [3]$. Let $E_{1,2} := \{\{u,v\} \in E : u \in V_1, v \in V_2\}$, and let $H := (V, E\backslash E_{1,2})$. Let $\mathbf{A} = \mathbf{A}_H$ denote the adjacency matrix of $H$ and let $\bar{\mathbf{A}}$ be a random signing of $\mathbf{A}$.

Suppose that $G$ has a triangle. Then, there exists a pair of vertices $u \in V_1, v \in V_2$ such that $\{u,v\} \in E_{12}$ and $H$ contains a path of length two between $u$ and $v$. Furthermore, observe that because there are no edges between $V_1$ and $V_2$ in $H$, $H$ has no paths of length three between $V_1$ and $V_2$. Consequently, in order to find a triangle in $G$, it suffices to check, for each $i \in V_1, j \in V_2$ with $\{i,j\} \in E_{1,2}$, whether there exists a path of length two between node $i$ and node $j$ in $H$. In other words, we need to check if there exists some $\{i,j\} \in E_{1,2}$ such that $\mathbf{A}^2_{i,j} > 0$. By Lemma 6, we can instead check if $\left|\bar{\mathbf{A}}^2_{i,j}\right| > 0$ and we would still correctly detect a triangle with probability at least 1/2. Note that this check requires requires only $O(\mathrm{nnz}(\mathbf{A}))$ additional time, since $|E_{1,2}| < \mathrm{nnz}(\mathbf{A})$.

So, our goal now is to compute an accurate enough estimate of $\left|\bar{\mathbf{A}}_{i,j}\right|^2$ for all $\{i,j\} \in E_{1,2}$ given the effective resistance estimate $\widetilde{r}_{i,j}$. To this end, let $\mathbf{N} = \left(\mathbf{I} - \frac{\alpha}{n}\bar{\mathbf{A}}\right)^{-1}$ for some $\alpha < \frac{1}{3}$. Note that the max row-sum of $\bar{\mathbf{A}}$ is 1/3, so the inverse exists. Lemma 11 guarantees that with high probability, $\rho\left(\bar{\mathbf{A}}\right) \leqslant \widetilde{O}(\sqrt{d_{\max}}) = \widetilde{O}(\sqrt{n})$. We condition on this event in the remainder of the proof. Consequently, we can express $\mathbf{N}$ as a power series,

$$\mathbf{N} = \sum_{k=0}^{\infty} \left(\frac{\alpha}{n}\right)^k \bar{\mathbf{A}}^k.$$

Now, let $\widetilde{\mathbf{N}}$ denote the truncation of $\mathbf{N}$ at the third term in this power series. That is, $\widetilde{\mathbf{N}} = \mathbf{I} + \frac{\alpha}{n}\bar{\mathbf{A}} + \frac{\alpha^2}{n^2}\bar{\mathbf{A}}^2$. Therefore, we have for all $\{i,j\} \in E_{1,2}$,

$$\bar{\mathbf{A}}^2_{i,j} = \frac{n^2}{\alpha^2}\widetilde{\mathbf{N}}_{i,j}.$$

Noticing that $\mathbf{N}_{i,j} = \frac{\mathbf{N}_{i,i}+\mathbf{N}_{j,j}-r_{i,j}}{2}$ motivates us to define our estimate of $\bar{\mathbf{A}}^2_{i,j}$ that we denote $P_{i,j}$ as follows

$$P_{i,j} := \frac{n^2}{\alpha^2}\left[\frac{\widetilde{\mathbf{N}}_{i,i} + \widetilde{\mathbf{N}}_{j,j} - \widetilde{r}_{i,j}}{2}\right].$$

Now, observe that $\widetilde{\mathbf{N}}_{i,i} = 1 + \frac{\alpha^2}{n^2}\bar{\mathbf{A}}^2_{i,i}$, and $\bar{\mathbf{A}}^2_{i,i}$ is simply the degree of vertex $i$ in $H$. Note that the random signing does not affect the fact that the diagonal entries of the square of the adjacency matrix are the degrees. Therefore, $\widetilde{\mathbf{N}}_{i,i}$ can also be computed for all $i$ in only $O(\mathrm{nnz}(\mathbf{A}))$ time. The additive error between our estimate $P_{i,j}$ and $\left|\bar{\mathbf{A}}\right|^2_{i,j}$ takes the form

$$\left|P_{i,j} - \bar{\mathbf{A}}^2_{i,j}\right| = \left|\frac{n^2}{\alpha^2}\left[\widetilde{\mathbf{N}}_{i,j} - \frac{\widetilde{\mathbf{N}}_{i,i} + \widetilde{\mathbf{N}}_{j,j} - \widetilde{r}_{i,j}}{2}\right]\right|.$$

By triangle inequality and plugging in the definition of $r_{i,j}$, we break up the error into four pieces

$$\left|P_{i,j} - \bar{\mathbf{A}}^2_{i,j}\right| \leqslant \frac{n^2}{\alpha^2}\left[\left|\widetilde{\mathbf{N}}_{i,j} - \mathbf{N}_{i,j}\right| + \left|\widetilde{\mathbf{N}}_{i,i} - \mathbf{N}_{i,i}\right|/2 + \left|\widetilde{\mathbf{N}}_{j,j} - \mathbf{N}_{j,j}\right|/2 + \left|\widetilde{r}_{i,j} - r_{i,j}\right|/2\right]. \quad (5)$$

We now bound each term separately. Consider any $i \in V_1$. Since, $H$ contains no triangle containing $i$, $\left(\bar{\mathbf{A}}^3\right)_{i,i} = 0$. So, we have that

$$\left|\mathbf{N}_{i,i} - \widetilde{\mathbf{N}}_{i,i}\right| = \left|\sum_{k=4}^{\infty}\frac{\alpha^k}{n^k}\left(\bar{\mathbf{A}}^k\right)_{i,i}\right| \leqslant \left\|\sum_{k=4}^{\infty}\frac{\alpha^k}{n^k}\bar{\mathbf{A}}^k\right\|_2$$

$$\leqslant \sum_{k=4}^{\infty} \widetilde{O}\left(\frac{\alpha}{\sqrt{n}}\right)^k = \frac{\widetilde{O}(\frac{\alpha}{\sqrt{n}})^4}{1 - \widetilde{O}(\frac{\alpha}{\sqrt{n}})}$$

Similarly, for any $i \in V_1$ and $j \in V_2$, note that $\bar{\mathbf{A}}_{i,j}^3 = 0$ because $H$ has no edges between $V_1$ and $V_2$, and there are no edges within each tripartition $V_i$. Consequently, by a similar argument as above,

$$\left| \mathbf{N}_{i,j} - \widetilde{\mathbf{N}}_{i,j} \right| = \sum_{k=4}^{\infty} (\bar{\mathbf{A}}^4)_{i,j} = \frac{1}{2} \sum_{k=4}^{\infty} e_i^\top (\bar{\mathbf{A}}^k) e_i + e_j^\top (\bar{\mathbf{A}}^k) e_j - \vec{\delta}_{i,j}^\top (\bar{\mathbf{A}}^k) \vec{\delta}_{i,j}$$

$$= \frac{1}{2} \left( e_i^\top e_i + e_j^\top e_j + \vec{\delta}_{i,j}^\top \vec{\delta}_{i,j} \right) \left\| \sum_{k=4}^{\infty} \frac{\alpha^k}{n^k} \bar{\mathbf{A}}^k \right\|_2$$

$$\leqslant 2 \sum_{k=4}^{\infty} \widetilde{O}\left(\frac{\alpha}{\sqrt{n}}\right)^k = \frac{2\widetilde{O}(\frac{\alpha}{\sqrt{n}})^4}{1 - \widetilde{O}(\frac{\alpha}{\sqrt{n}})}.$$

Finally, consider the magnitude of the approximation error between $\widetilde{r}_{i,j}$ and $r_{i,j}$. We have

$$|r_{i,j} - \widetilde{r}_{i,j}| \leqslant \epsilon |r_{i,j}|.$$

Note that $|r_{i,j}| = \left| \vec{\delta}_{i,j}^\top \mathbf{N} \vec{\delta}_{i,j} \right| \leqslant \|\vec{\delta}_{i,j}\|^2 \|\mathbf{N}\| \leqslant 2\| \left(\mathbf{I} - \frac{\alpha}{n}\bar{\mathbf{A}}\right)^{-1} \| \leqslant \frac{2}{1 - \frac{\alpha}{n}\|\bar{\mathbf{A}}\|_2} \leqslant \frac{2}{1 - \widetilde{O}(\frac{\alpha}{\sqrt{n}})}$. Plugging these estimates of the errors into (5), we get

$$\left| P_{i,j} - \bar{\mathbf{A}}_{i,j}^2 \right| \leqslant \frac{n^2}{\alpha^2} \left[ \frac{\widetilde{O}\left(\frac{\alpha}{\sqrt{n}}\right)^4 + \epsilon}{1 - \widetilde{O}\left(\frac{\alpha}{\sqrt{n}}\right)} \right].$$

Since $\left| \bar{\mathbf{A}}_{i,j}^2 \right| \in \{0, 1\}$, to compute $\bar{\mathbf{A}}_{i,j}^2$, we need approximate it to additive $1/2$ error. That is, we require

$$\frac{1}{1 - \widetilde{O}(\frac{\alpha}{\sqrt{n}})} \left[ \widetilde{O}(\alpha^2) + \epsilon \frac{n^2}{\alpha^2} \right] < \frac{1}{2},$$

or equivalently,

$$\epsilon < \frac{\alpha^2 - \widetilde{O}(\frac{\alpha^3}{\sqrt{n}})}{2n^2} - \frac{\widetilde{O}(\alpha^4)}{n^2} = \widetilde{O}\left(\frac{1}{n^2}\right),$$

where the last step follows from the fact that we can take $\alpha$ to be a sufficiently small constant. Therefore, estimates $\widetilde{r}_{i,j}$ with $\epsilon = \widetilde{O}(n^{-2})$ for all $\{i,j\} \in E_{1,2}$ are sufficient to determine if $\left| \bar{\mathbf{A}}_{i,j}^2 \right|$ is 0 or 1. As noted earlier, checking this for all edges in $E_{1,2}$ takes $O(\mathrm{nnz}(\mathbf{A}))$ additional time. Therefore, by plugging in $\epsilon = \widetilde{O}(n^{-2})$, we can use an algorithm that solves the SDD effective resistance estimation problem in $\widetilde{O}(n^2 \epsilon^{-c})$ time to solve the triangle detection problem in $\widetilde{O}(n^2 n^{2c}) = \widetilde{O}(n^{2(1+c)})$ time whp. and this completes the proof. $\qquad\square$

**Lemma 7.** *Given an algorithm to solve the all edges effective resistance estimation problem (i.e., Definition 1 where $S = E$) in $\widetilde{O}(m \epsilon^{-c})$ time, we can produce an algorithm to solve the all pairs effective resistance estimation problem in $\widetilde{O}(n^2 \epsilon^{-c})$ time for some $c > 0$.*

*Proof.* The idea is that, given a graph $G$ on $n$ nodes and $m$ edges, we can always add to it a complete graph of edges of sufficiently small weight that would not change the effective resistances much. Let $\mathbf{L}_G$ be the graph Laplacian and $H$ be the graph obtained by adding a complete graph with uniform edge weight $\alpha > 0$. Then, $\mathbf{L}_H = \mathbf{L}_G + \alpha(n\mathbf{I} - \mathbb{1}\mathbb{1}^T)$. It suffices to find $\alpha$ small enough such that $x^T \mathbf{L}_H^\dagger x \approx_\epsilon x^T \mathbf{L}_G^\dagger x$ for all $x \perp \mathbb{1}$. First, we use the fact that if $x^T \mathbf{L}_H x \approx_\epsilon x^T \mathbf{L}_G x$, then we also have $x^T \mathbf{L}_H^\dagger x \approx_\epsilon x^T \mathbf{L}_G^\dagger x$ [28]. To show the former, we need

$$x^T (\mathbf{L}_G + \alpha(n\mathbf{I} - \mathbb{1}\mathbb{1}^T) - \mathbf{L}_G)x \leqslant \epsilon x^T \mathbf{L}_G x$$

$$\implies \alpha \leqslant \epsilon \frac{x^T \mathbf{L}_G x}{x^T (n\mathbf{I} - \mathbb{1}\mathbb{1}^T) x}$$

for all $x \perp \mathbb{1}$. Therefore, taking $\alpha \leqslant \epsilon \frac{\lambda_{\min}(\mathbf{L}_G)}{n}$ is sufficient to have $x^T \mathbf{L}_H^\dagger x \approx_\epsilon x^T \mathbf{L}_G^\dagger x$ for all $x \perp \mathbb{1}$. As a consequence, we can estimate all pairs effective resistances on $G$ by estimating all edges effective resistances on $H$. So, any combinatorial $\tilde{O}(m\epsilon^{-c})$ time algorithm for the all edges effective resistance estimation problem would immediately imply a combinatorial $\tilde{O}(n^2\epsilon^{-c})$ time algorithm for all pairs effective resistance estimation. $\qquad\square$

### 6.3.2 Proofs for Lower Bounds on Spectral Sum Estimation

In this section, we present proofs for the results presented in Section 4.3. First, we prove Lemma 8 below.

**Lemma 8.** *If* $\mathrm{tr}\left(\mathbf{A}_G^3\right) = 0$, *then* $\mathrm{tr}\left(\bar{\mathbf{A}}_G^3\right) = 0$, *and if* $\mathrm{tr}\left(\mathbf{A}_G^3\right) > 0$ *then* $\mathbb{P}\left[\left|\mathrm{tr}\left(\bar{\mathbf{A}}_G^3\right)\right| > 0\right] \geqslant 1/4$.

*Proof.* Again for convenience, let $\bar{\mathbf{A}} = \bar{\mathbf{A}}_G$. Denote by $\xi_{ab} = \xi_{ba}$ the Rademacher random variable used to decide the sign of edge $(a, b)$. We can write

$$\mathrm{tr}\left(\bar{\mathbf{A}}^3\right)/6 = \sum_{\text{triangles } \{i,j,k\} \text{ in } G} \xi_{ij}\xi_{jk}\xi_{ki} =: T \tag{6}$$

If $\mathrm{tr}\left(\mathbf{A}^3\right) > 0$, then $G$ must have at least one triangle. Consider the following cases:

1. $G$ has an odd number of triangles. In this case, since each term in the sum in (6) is either $+1$ or $-1$, and there is an odd number of terms in the sum, so $\mathrm{tr}\left(\bar{\mathbf{A}}^3\right) > 0$ wp 1.

2. $G$ has an even number of triangles. First, define $T(a,b) :=$ $\sum_{\text{triangles } \{a,b,k\} \text{ that contain } \{a,b\}} \xi_{ab}\xi_{bk}\xi_{ak}$ We now subdivide this into two cases:

   (a) There exists an edge $\{a, b\}$ that is part of an odd number of triangles. We can decompose the sum in (6) as follows:

   $$\mathrm{tr}\left(\bar{\mathbf{A}}^3\right)/6 = \underbrace{T(a,b)}_{S_1} + \underbrace{T - T(a,b)}_{S_2}$$

   Suppose there exists some realization of the random variables $\xi_{ij}$ such that $\mathrm{tr}\left(\bar{\mathbf{A}}^3\right)/6 = 0$. Since $S_1$ has odd terms, it must be non-zero. By flipping the sign of $\xi_{ab}$, we can flip the sign of $S_1$, and so $-S_1 + S_2 \neq 0$. Therefore, for every configuration the variables $\xi_{ij}$ that result in a 0 trace, there exists an equally likely configuration that results in $\mathrm{tr}\left(\bar{\mathbf{A}}^3\right)/6 \neq 0$. Therefore, $\mathbb{P}\left[\left|\mathrm{tr}\left(\bar{\mathbf{A}}^3\right)\right| > 0\right] \geqslant 1/2$.

   (b) Every edge in $G$ is a part of an even number of triangles. Let $\{a, b, c\}$ be a triangle in $G$. In this case, we decompose (6) as follows:

   $$\mathrm{tr}\left(\bar{\mathbf{A}}^3\right)/6 = \underbrace{\xi_{ab}\xi_{bc}\xi_{ac}}_{S_1} + \underbrace{T(a,b) - \xi_{ab}\xi_{bc}\xi_{ac}}_{S_2} + \underbrace{T(b,c) - \xi_{ab}\xi_{bc}\xi_{ac}}_{S_3}$$
   $$+ \underbrace{T - T(a,b) - T(b,c) + \xi_{ab}\xi_{bc}\xi_{ac}}_{S_4}$$

   Consider all 4 possible values of the pair of random variables $\{\xi_{ab}, \xi_{bc}\}$. Since each $S_i$ has an odd number of terms, $S_i \neq 0$ for all $i$. We observe that it is not possible for all 4 equally likely configurations of $\{\xi_{ab}, \xi_{bc}\}$ to result in $S_1 + S_2 + S_3 + S_4 = 0$, so at least one configuration must result in $T \neq 0$. Therefore, $\mathbb{P}\left[\left|\mathrm{tr}\left(\bar{\mathbf{A}}^3\right)\right| > 0\right] \geqslant 1/4$.

$\qquad\square$

To prove our main hardness result Theorem 5, we first present a hardness result for a more general class of spectral sums in the theorem below. It closely resembles Theorem 15 from [33], but with improved bounds as a result of the random signing.

**Theorem 11** (Improved randomized version of Theorem 15 from [33]). *Let $f : \mathbb{R}^+ \to \mathbb{R}^+$ be a function such that it can be expressed as $f(x) = \sum_{k=0}^{\infty} c_k(x-1)^k$ where $|c_k/c_3| \leqslant h^{k-3}$ for $k > 3$ and $x \in (0, 2)$. Given an algorithm which takes as input a graph $G = (V, E)$ on $n$ nodes and, in $O(n^\gamma \epsilon^{-c})$ time, outputs an estimate $X \approx_{\epsilon_1/9} \mathcal{S}_f(\mathbf{I} - \delta \bar{\mathbf{A}}_G)$ with $\delta$ and $\epsilon_1$ satisfying $\delta = \min\{(\sqrt{n}\log(\alpha n))^{-1}, (10n^3 h \log(\alpha n))^{-1}\}$ and $\epsilon_1 = \min\{1, |c_3\delta^3/(c_0 n)|, |c_3\delta/(c_2 n^2)|\}$ for some constant $\alpha > 1$, we can produce an algorithm that solves the triangle detection problem in $O(n^2 + n^\gamma \epsilon_1^{-c})$ time whp.*

*Proof.* For convenience, we write $\mathbf{A} = \mathbf{A}_G$. The proof closely follows the proof of Theorem 15 from [33], but we replace $\mathbf{A}$ with its symmetrically random signed version that we denote $\bar{\mathbf{A}}$. We present a full proof here for completeness.

First, we note that by lemma 11, we have that $\|\bar{\mathbf{A}}\|_2 \leqslant \sqrt{n}\log\alpha n$ whp. for some constant $\alpha > 1$. We define $\bar{\mathbf{B}} = \mathbf{I} - \delta\bar{\mathbf{A}}$ and consequently $\bar{\mathbf{B}}$ is PSD whp. Now, using the definition of $f$, we have

$$\sum_{i=1}^{n} \sigma_i(\bar{\mathbf{B}}) = \sum_{i=1}^{n} f(1 - \delta\lambda_i(\bar{\mathbf{A}})) = \sum_{i=1}^{n}\sum_{k=0}^{\infty} c_k(\delta\lambda_i(\bar{\mathbf{A}}))^k = \sum_{k=0}^{\infty} c_k\delta^k \text{tr}\left(\bar{\mathbf{A}}^k\right).$$

We analyze the tail of this power series. Specifically, we have

$$\left|\sum_{k=4}^{\infty} c_k\delta^k \text{tr}\left(\bar{\mathbf{A}}^k\right)\right| \leqslant |c_3|\,\delta^3 \sum_{k=4}^{\infty} \left|\text{tr}\left(\bar{\mathbf{A}}^k\right)\right|\delta^{k-3}\left|\frac{c_k}{c_3}\right|. \tag{7}$$

Now, we have $\left|\text{tr}\left(\bar{\mathbf{A}}^k\right)\right| \leqslant \|\bar{\mathbf{A}}\|_2^{k-2}\|\bar{\mathbf{A}}\|_F \leqslant n^{k/2+1}$ whp. and further $n^{k/2+1} \leqslant n^{3(k-3)}$ for all $k > 3$. Therefore, using the definition of $\delta$ as in the theorem we get that whp.,

$$\left|\text{tr}\left(\bar{\mathbf{A}}^k\right)\right|\delta^{k-3}\left|\frac{c_k}{c_3}\right| \leqslant \frac{1}{10^{k-3}} \text{ for all } k > 3.$$

Plugging into Equation (7), we get

$$\left|\sum_{k=4}^{\infty} c_k\delta^k \text{tr}\left(\bar{\mathbf{A}}^k\right)\right| \leqslant \frac{|c_3|\,\delta^3}{9}.$$

The rest of the proof is essentially identical to the steps in the proof of Theorem 15 in [33], but we reproduce them here for completeness.

Using the simple facts $\text{tr}\left(\bar{\mathbf{A}}^0\right) = n$, $\text{tr}\left(\bar{\mathbf{A}}\right) = 0$ and $\text{tr}\left(\bar{\mathbf{A}}^2\right) \leqslant n^2$, we have

$$c_0\text{tr}\left(\bar{\mathbf{A}}^0\right) + c_1\text{tr}\left(\bar{\mathbf{A}}\right) + c_2\text{tr}\left(\bar{\mathbf{A}}^2\right) \leqslant |c_3|\,\delta^3\left(c_0 n/(c_3\delta^3) + c_2 n^2/(c_3\delta)\right) \leqslant \frac{|c_3|\,\delta^3}{\epsilon_1}.$$

Given $X$, in $O(\text{nnz}(\bar{\mathbf{A}}))$ time, we can compute

$$X - c_0 n - c_2\delta^2\text{tr}\left(\bar{\mathbf{A}}^2\right) = c_3\delta^3\text{tr}\left(\bar{\mathbf{A}}^3\right) \pm \frac{|c_3|\,\delta^3}{9} \pm \frac{\epsilon_1}{9}\left(\frac{|c_3|\,\delta^3}{9} + c_3\delta^3\text{tr}\left(\bar{\mathbf{A}}^3\right) + \frac{|c_3|\,\delta^3}{\epsilon_1}\right)$$

$$= c_3\delta^3\left[\text{tr}\left(\bar{\mathbf{A}}^3\right)\left(1 \pm \frac{1}{20}\right) \pm \frac{1}{3}\right].$$

This is sufficient to detect if $\left|\text{tr}\left(\bar{\mathbf{A}}\right)\right| = 0$ or if $\left|\text{tr}\left(\bar{\mathbf{A}}^3\right)\right| \geqslant 1$. The final result then follows by applying Lemma 8. $\qquad\square$

We now prove the main result Theorem 5.

**Theorem 5.** *Given a combinatorial algorithm which on input $\mathbf{B} \in \mathbb{R}^{n \times n}$ outputs a spectral sum estimate $Y \approx_\epsilon \mathcal{S}_f(\mathbf{B})$ in $O(n^\gamma \epsilon^{-c})$ time with $\gamma \geqslant 2$ for the spectral sums in Table 3, we can produce a randomized combinatorial algorithm that can detect a triangle in an n-node graph whp. in $\widetilde{O}(n^{\gamma+\alpha c})$ time, where $\alpha$ is a scaling that depends on properties of the function $f$ (see Table 3 for values of $\alpha$ for several spectral sums.)*

*Proof.* We apply Theorem 11 to the specific spectral sums in Table 3.

**Schatten 3-norm** We have $f(x) = x^3$. Therefore, $c_k = 0$ for $k > 3$. So we apply Theorem 11 with $h = 0$ and hence $\delta = \widetilde{O}(\frac{1}{\sqrt{n}})$ and $\epsilon_1 = \widetilde{O}(\frac{1}{n^{2.5}})$.

**Schatten p-norm** $p \neq 1, 2$ We have $f(x) = x^p$. Using the Taylor series about 1, we have $\frac{c_k}{c_3} \leqslant p^{k-3}$ for all $k > 3$ as well as $\left|\frac{c_0}{c_3}\right| = \left|\frac{1}{p(p-1)(p-2)}\right| \leqslant \left|\frac{1}{2\min\{p,(p-1),(p-2)\}}\right|$ and similarly $\left|\frac{c_2}{c_3}\right| \leqslant \left|\frac{1}{2\min\{p,(p-1)\}}\right|$. Therefore, with $h = p$, we apply Theorem 11 with $\delta = \widetilde{O}(\frac{1}{n^3 p})$ and $\epsilon_1 = \frac{c_3 \delta^3}{c_0 n} = \widetilde{O}(\frac{|\min\{p,(p-1),(p-2)\}|}{n^{10} p^3})$, which gives the result.

**SVD Entropy** We have $f(x) = x \log x$. For $x \in (0, 2)$, using the Taylor Series about 1 we can write $x \log x = \sum_{k=0}^{\infty} c_k (x-1)^k$ where $c_0 = 1 \log(1) = 0$, $c_1 = \log(1) + 1 = 1$, and $|c_k| = \frac{(k-2)!}{k!} \leqslant 1$ for $k \geqslant 2$. So we have $c_k < c_3$ for all $k > 3$, $\frac{c_0}{c_3} = 0$ and $\frac{c_2}{c_3} = \frac{1}{3}$. So with $h = 1$, Applying Theorem 11 with $\delta = \widetilde{O}(\frac{1}{n^3})$ and $\epsilon_1 = \frac{\delta}{3n^2} = \widetilde{O}(\frac{1}{n^5})$ gives the result.

**Log Determinant** We have $f(x) = \log x$. For $x \in (0, 2)$, using the Taylor Series about 1 we can write $\log x = \sum_{k=0}^{\infty} c_k (x-1)^k$ where $c_0 = 0$, $|c_i| = 1/i$ for $i \geqslant 1$. Again we have $c_k < c_3$ for all $k > 3$ and $\frac{c_0}{c_3} = 0$ while $\frac{c_2}{c_3} = \frac{3}{2}$. So with $h = 1$, Applying Theorem 11 with $\delta = \widetilde{O}(\frac{1}{n^3})$ and $\epsilon_1 = \frac{\delta}{3n^2} = \widetilde{O}(\frac{1}{n^5})$ gives the result.

**Trace of Exponential** We have $f(x) = e^x$. Using the Taylor Series about 1 we can write $e^x = \sum_{k=0}^{\infty} \frac{e(x-1)^k}{k!}$. We have $\frac{c_0}{c_3} = 6$, $\frac{c_2}{c_3} = 3$, and $c_k < c_3$ for all $k \geqslant 3$. So with $h = 1$, Applying Theorem 11 with $\delta = \widetilde{O}(\frac{1}{n^3})$ and $\epsilon_1 = \frac{c_3 \delta^3}{c_0 n} = \widetilde{O}(\frac{1}{n^{10}})$ gives the result. $\qquad\square$