# OpenReview forum: "Towards Optimal Effective Resistance Estimation"
_NeurIPS.cc/2023/Conference — NeurIPS 2023 poster_

### Official Review · Reviewer_UUSU · 2023-06-29

**Soundness:** 4 excellent
**Presentation:** 3 good
**Contribution:** 2 fair
**Rating:** 6
**Confidence:** 3

**Summary:**

The authors consider the problem of estimating effective resistance both for all edges of a graph and between any pair of nodes. This quantity is incredibly useful for sampling from a graph to produce a sparser graph with similar spectral properties. Furthermore, this quantity can provide a useful metric for the importance of an edge to the connectivity of the network. Within this context the authors restrict to a subclass, expander graphs, which have better connectivity properties. The practical application of these graphs is much more limited, but as the authors point out, they are commonly used as a first step towards giving more general algorithms. The authors provide a new algorithm that for sparse graphs gives an asymptotically faster algorithm than the previous work. In addition, the authors improve upon the lower bound runtime by reducing to the triangle detection problem.

**Strengths:**

1. Improves the asymptotic runtime for an important problem on sparse graphs of a commonly studied subclass, expander graphs.

2. Applies a new clever factorization technique for expander graphs with substantial technical depth

3. Gives a new approach for lower bounding the runtime through a reduction to the triangle detection problem that extends to additional problems

**Weaknesses:**

For a broad ML venue the practical application of this theoretical work is quite limited given the restriction to expander graphs. It makes good progress in a well-studied area in the theoretical community but may be more appropriate for a theory-centric venue. To be of more general interest, it would be preferable to see progress upon general graphs by either making asymptotic or practical improvements. I would very much welcome the authors rebuttal (see questions section) but it also feels like these techniques, specifically the asymmetric factorization bounding, which is very clever, could not be reasonably extended to general graphs.

**Questions:**

What's the biggest bottleneck in your opinion for applying similar techniques to the ones you provide in this work to improve the runtime for general graphs?

**Limitations:**

The authors adequately addressed the limitations

---

> ### Author Rebuttal · Authors · 2023-08-09
>
> Thank you for your thoughtful comments and feedback! We have addressed the limitations of the expander case and high-level motivation for considering the general response (see Comment 3 therein). In addition, we have described the key bottleneck in the general response as well. Thank you again for your suggestions!

---

> > ### Comment · Reviewer_UUSU · 2023-08-10
> >
> > Thanks for all the detailed responses! I appreciate the authors pointing out that their asymmetric factorization bound does apply (just not as well) to general graphs which I had overlooked in my review. It's always difficult to conjecture how well these techniques could lead to further improvement for general graphs. But I'm curious if the authors have any opinion on the most likely avenue for achieving nearly linear time on general graphs (improving asymmetric factorization bound or applying expander decomposition techniques that they mention in the rebuttal).
> >
> > My intuition is still is that improving their asymmetric factorization bound will be more difficult than figuring out a different approach for general graphs. But regardless, the authors have several nice results that warrant acceptance in my opinion.

---

> > > ### Author Response · Authors · 2023-08-11
> > >
> > > Thanks for the question! As you already mentioned (and we discussed in our general response), it is challenging to conjecture how to obtain the runtime for general graphs. (Though this is certainly an interesting open problem for future work.)
> > >
> > > It’s unclear how to improve the asymmetric factorization bound _directly_  and we are unsure whether a better direct bound will be possible. One approach for leveraging our results in future work is to use algorithms on expanders (such as ours, or others) as a subroutine in a general graph algorithm in order to get improved runtimes for effective resistance estimation, or at least better query times for sketching. This might include (1) the technique of repeatedly computing effective resistances in expander decompositions/partitions (like what is done [2] and [3]) and (2) a recursive Gaussian elimination approach like what is also used in [3]. However this is speculative and for future research.

---

> > > > ### Comment · Reviewer_UUSU · 2023-08-14
> > > >
> > > > Thanks for the response! I appreciate the authors willingness to speculate on future directions of their work, and their approaches mentioned seem reasonable. I especially appreciate the authors speculating upon improving the asymmetric factorization bound given that they have the best intuition upon this technique and the potential for extending it.

---

### Official Review · Reviewer_AnJp · 2023-07-04

**Soundness:** 3 good
**Presentation:** 3 good
**Contribution:** 3 good
**Rating:** 6
**Confidence:** 2

**Summary:**

The paper studies the (approximate) estimation of effective resistance on expander graphs, i.e., graphs with edge expansion $\Omega(1/\text{polylog}\ n)$. The main upper-bound result is a data structure (sketch) for effective resistance that takes $\tilde{O}(m/\epsilon)$ time to build, answers each query in $\tilde{O}(1)$ time, and is of size $\tilde{O}(m/\epsilon)$. This improves the Li and Sachdeva [SODA’23] on the construction time on sparse expanders with $O(n/\epsilon)$ edges. The data structure implies an $\tilde{O}(n^2/\epsilon)$-time algorithm to compute all-pairs effective resistance. To complement the upper bound result, the paper further includes a lower bound of $\tilde{\Omega}(n^2/\epsilon^{0.5})$ time for computing all-pairs effective resistance for all algorithms that are not based on fast matrix multiplications (FMM).

The main technical ingredient for the upper bound result is to use asymmetric factorization of the effective resistance in expander graphs (in the form of $\ell_{1}$ norm) and apply the count-sketch in $\ell_{1}$ to achieve the $\epsilon^{-1}$ compression. For the lower bound side, the paper designs a novel reduction to the triangle detection problem, which improves the exponent on $\epsilon$ from $1/13$ to $1/2$.

My general evaluation of the paper is positive. The paper seems to require a good deal of background to read, and I did not manage to check the correctness of the proofs. The high-level intuitions make sense to me, and conditioning on equation (2) is correct, I think everything goes through for the upper bound. As such, as far as I can see, the paper is technically solid, and it makes progress toward an interesting open problem. On the flip side, it seems the improvements of this paper are rather restrictive – compared to Li and Sachdeva [SODA’23], it is only better on expander with $O(n/\epsilon)$ edges. For the presentations, I think the paper generally makes good efforts to define notions and explain ideas clearly. However, I still believe that the writing assumes a very strong level of knowledge from the reader (see the weakness part for details).


**Strengths:**

As I have mentioned in the general evaluation, I think the paper is technically solid, and the presentation is clear in general. The high-level intuitions provided in the paper are helpful to the readers, and the techniques include novel ideas of the symmetric factorization of the effective resistance.


**Weaknesses:**

Most of them are already mentioned in the general evaluation. Here is a list that would help the authors to address:
- On the upper bound side, the improvement compared to Li and Sachdeva [SODA’23] is rather limited – the bound is only better on expanders with $O(n/\epsilon)$ edges.
- As I said in the general evaluation, the paper seems to require a good deal of background to read. I realize that the authors have already put efforts into making the paper more accessible by defining most of the notions clearly from the get-go and presenting the high-level ideas concisely. However, in my (very subjective) opinion, reading the paper still requires some non-trivial efforts.
- I think some of the presentation problems can be attributed to unclear notations, for instance:
1. $\tilde{O}(\cdot)$ is defined after multiple usages in section 1, and $\tilde{\Omega}(\cdot)$ is never defined. This is especially confusing since I think you hide $\text{polylog}$ vs. $1/\text{polylog}$ factors in the two notations.
2. The edge-vertex incident matrix $B_{G}$ is briefly mentioned but never formally defined (I know that it is): this could be an issue for the general audience at NeurIPS to understand.
3. Is $J$ in line 194 the matrix for the JL projection? It is never mentioned.
- Finally, for NeurIPS, it would be good if you could spend more passage talking about downstream applications and what your bound would imply therein.


**Questions:**

Most of them are presentation-related issues, and some of them are comments
- Line 63, do you mean $(\tilde{O}(m \epsilon^{-1}), \tilde{O}(1), \tilde{O}(n \epsilon^{-1}))$? (The current notation means sublinear time but linear size.)
- Similarly, the order seems to be wrong in line 187 – should be $(\tilde{O}(m \epsilon^{-2}), \tilde{O}(\epsilon^{-2}), \tilde{O}(n \epsilon^{-2}))$ according to your notation.
- Line 120: ‘Building on this work’ – what do you mean? Based on this manuscript?
- Line 198: there are two ‘vectors’.
- Formatting issue: For some reason, most of your citations do not contain the venue of publication; some of them have the venue but not the title – please fix this.
- Minor formatting issue: adding *actual citations* in the abstract seems to be non-standard, i.e., I think it is ok to write as (Chu et al. [Arxiv’18]), but not putting the actual pointer there.
- Did you state the (conditional) hardness result for triangle detection anywhere in the paper? I would recommend including it in the preliminary.
- I do not quite understand the logic of ruling out FFM-based algorithms: from the statement of theorem 3, it seems the reduction between effective resistance and triangle detection is unconditional. So why the reduction does not work for FFM-based algorithms? Is it because of the hardness result of triangle detection?


**Limitations:**

I do not see any immediate non-technical/societal limitations on the work. Presentation-wise, it would be good if the paper had discussed more downstream applications.

---

> ### Author Rebuttal · Authors · 2023-08-09
>
> Thank you for your thoughtful comments and questions! In the general response, we have discussed how we plan to incorporate some of the changes to the preliminary/introduction of the paper in response to your helpful suggestions for content to include therein.
>
> **Question 1: Clarifying question on Line 120. What do you mean by “building on this work”?**
>
> **Response:** Thank you for the question, and we apologize for the confusion. “Building on this work” was intended to “Building on this line of research…” We did not mean to imply that [1] builds on our work. Instead, we meant that [1] builds on the aforementioned long line of research on sketching and sparsification. We will update the language to “Building on the aforementioned line of research on graph sketching and sparsification, …” to improve clarity.
>
> **Question 2: There is a formatting issue in the bibliography where venues are not showing up properly.**
>
> **Response:** Thank you for pointing this out; we will update the references to include the venue in the next iteration of the submission. Sorry for any inconvenience this caused.
>
> **Question 3: There are typos on lines 63, 187, 120, and 198.**
>
> **Response:** Thank you for catching these typos. It should indeed be $\tilde{O}(m/\epsilon), \tilde{O}(1), \tilde{O}(n/\epsilon)$ on Line 63, and $(\tilde{O}(m \epsilon^{-1}), \tilde{O}(\epsilon^{-2}), \tilde{O}(n\epsilon^{-2})$ on Line 187. Thank you for catching the duplicated word vector on line 198. These errors will be corrected in the next iteration of the submission.
>
> **Question 4: The triangle detection hardness result is not stated.**
>
> **Response:** We discuss the subcubic triangle detection problem and its conjectured hardness in line 70, in the body, but we did not state it formally inside a definition environment. We will update this in the next iteration of the submission to state the problem formally as well as concrete conjecture in a separate environment to improve the clarity.
>
> **Question 5: The abstract citations are nonstandard.**
>
> **Response:** Thank you for the feedback, we will modify our abstract towards your suggested form of citations.
>
> **Question 6: I don't quite understand the logic of ruling out FMM-based algorithms, as Theorem 3 does not say anything about FMM.**
>
> **Response:** Thank you for the question. Our reduction from algorithms for effective resistance to algorithms for triangle detection actually works for any type of algorithm for effective resistance. This is why our statement of Theorem 3 does not mention FMM. Informally, Theorem 3 essentially says that if we have some type of effective resistance algorithm, we can produce a similar type of triangle detection algorithm. **One** implication of Theorem 3 is the following: if we have a combinatorial algorithm for effective resistance (i.e., non-FMM based algorithm) with some runtime then we can get a combinatorial algorithm for triangle detection with a related runtime. We elaborate on this below.
>
> Although the reduction stated in Theorem 3 does not make any assumption on the type of effective resistance algorithm we start with, Theorem 3 only implies a conditional lower bound against non-FMM based (i.e., combinatorial) algorithms for effective resistance (barring a major advance in combinatorial algorithms). This is because there already exists a simple subcubic algorithm for triangle detection using FMM (we can take the adjacency matrix A raised to the power of 3, and check whether any of the diagonal entries of A^3 are positive.) The **conditional hardness stems from the fundamental challenge** of obtaining subcubic triangle detection algorithms which do not rely on FMM. Obtaining any non-FMM algorithm for triangle detection would be a major algorithmic breakthrough, and consequently, it is a common hardness assumption in the literature. Section 6.1 of the supplementary material points to a few relevant reference.
>
> **Question 7: It would be great if the authors can discuss more downstream applications.**
>
> **Response:** Thank you for your suggestion. We discuss some downstream applications of effective resistance estimation more broadly in the introduction (Line 23-37), but we would be open to include more discussions, perhaps in the conclusion or elsewhere if you feel it would be helpful. Please let us know if you have any recommendations for specific applications or discussions you would like for us to incorporate.

---

> > ### Comment · Reviewer_AnJp · 2023-08-13
> > **Reponse**
> >
> > Thanks for responding to my questions. I think many clarifications in the responses are worth adding to the paper to help readers understand. Please take advantage of the extra page in the final version to include these. Also, it'll be helpful to add some aspects of these discussions to a full version of your paper.

---

> > > ### Author Response · Authors · 2023-08-16
> > >
> > > Thank you for the feedback! We indeed plan to incorporate many of the clarifications discussed in the author-reviewer discussions in the next iteration of the submission. Thanks!

---

### Official Review · Reviewer_9w9Q · 2023-07-05

**Soundness:** 3 good
**Presentation:** 3 good
**Contribution:** 4 excellent
**Rating:** 7
**Confidence:** 4

**Summary:**

In the paper, the authors study the problem of estimating the effective resistance between pairs of vertices in weighted, undirected expander graphs. The authors present a novel algorithm for effective resistance estimation and effective resistance sketching. Moreover, they give a conditional lower bound based on triangle detection.

**Strengths:**

This is a very nice theory paper that studies the problem of effective resistance estimation which a lot of people in the relevant community care.  The contributions of the paper are novel algorithms for both the lower bound and upper bound. The algorithm cleverly uses the countSketch to build a spectral sketch data structure and connects the Laplacian of expanders using Cheeger's inequality. The paper is very well written and it is a pleasure to read. In our opinion, this is an important contribution to the literature.



**Weaknesses:**

For the upper bound,  the result looks very similar to the result of [Li, Sachdeva 2022]. It would be great if the authors can provide a more close comparison between the two.


**Questions:**


Minor:
1. Line 63: should be $(\widetilde{O}({m\epsilon^{-1}}),\widetilde{O}(1), \widetilde{O}({n\epsilon^{-1}}))$-effective
2. Line 192: the expression of $r_G(i,j)$ missing the $\delta_{i,j}$ term
3. Both Equation (1) and Equation (2) missing the $\delta_{i,j}$ term
4: Line 240: peridicular should be perpendicular?

---

> ### Author Rebuttal · Authors · 2023-08-09
>
> Thank you for your thoughtful comments and questions!
>
> **Question 1: It would be helpful if the authors could provide a more close comparison between their work and that of [3].**
>
> **Response:** Thanks for the suggestion. We would be happy to incorporate more of this comparison in the introduction of the paper in the next iteration of our submission. Due to space limitations, we deferred most of this discussion to the supplementary material (beginning on line 536 in Section 6.1). In addition to the comparison in the supplementary material, we would like to highlight the following differences:
> * For expanders, in the regime where $m << n\epsilon^{-1}$, i.e., for sparse enough graphs, our runtime is faster than that of [3]. In the dense regime where $m >> n\epsilon^{-1}$, our result can immediately be improved to an $\tilde{O}(m^{1 + o(1)} + n\epsilon^{-2})$ runtime algorithm, by running on a standard sparse sketch of the original graph (following the standard approach around Line 118 of the introduction). Consequently, our result improves over that of [3] for sufficiently sparse graphs, and almost matches their result (up to an $m^{o(1)}$) factor in the dense regime.
> * For general graphs, we also obtain a runtime of $\tilde{O}(m\bar{\kappa}(L_G) \epsilon^{-1})$, while [3] obtain a runtime with a worse dependence on $\bar{\kappa}(L_G)$. In this sense, our algorithms are less sensitive to the conditioning of the graph than those of [3].
>
> **Question 2: There is a typo on line 63.**
>
> **Response:** It should indeed be $\tilde{O}(m/\epsilon), \tilde{O}(1), \tilde{O}(n/\epsilon)$ on Line 63 and will be fixed in the next iteration. Thanks!
>
> **Question 3: There is a typo on line 192.**
>
> **Response:** The $\delta_{i,j}$ was indeed missing in Line 192, as well as in the second term of Equation (1) and (2). This will be fixed in the next iteration of the submission. Thanks!
>
> **Question 4: Perpendicular is miss-spelled in Line 240.**
>
> **Response:** Thanks! This will also be fixed in the next iteration of the submission.

---

> > ### Comment · Reviewer_9w9Q · 2023-08-14
> >
> > Thank you for your detailed response.

---

### Official Review · Reviewer_btut · 2023-07-10

**Soundness:** 4 excellent
**Presentation:** 4 excellent
**Contribution:** 4 excellent
**Rating:** 8
**Confidence:** 4

**Summary:**

The paper theoretically analyzes the time complexity of estimating effective resistances in $n$-node $m$-edge undirected, expander graphs. Specifically, the paper present an algorithm which can achieve $\tilde{O}(m \varepsilon^{-1})$ time complexity for estimating the effective resistance of all edges in such graphs. This complexity bound significantly improves the previous result either for general graphs or for expanders. The paper also present a tight lower bound for computing the estimates of effective resistances.

**Strengths:**

S1. Computing the effective resistance between a pair of graph nodes is an important and fundamental problem. The paper improves the best complexity result of existing work with non-trivial analysis. A tight lower bound is also provided in the paper.

S2. The paper's technical part is sound and solid.

S3. The paper is well-written and easy to follow.

**Weaknesses:**

W1. The paper lacks experimental results to empirically demonstrate the effectiveness of the proposed algorithm.


**Questions:**

The paper mainly focuses on undirected expander graphs. An interesting question is whether the complexity results given in the paper can be further extended to general graphs.

**Limitations:**

I do not see any potential negative societal impact in this paper.

---

> ### Author Rebuttal · Authors · 2023-08-09
>
> Thank you for your thoughtful comments and questions!
>
> We have discussed interesting directions for future empirical experiments in the general response, as it was a common theme brought up by multiple reviewers. In addition, we have discussed the primary bottleneck to extending results to general graphs in the general response. Please let us know if you have any additional feedback on these points, and we thank you again for your review.

---

> > ### Comment · Reviewer_btut · 2023-08-14
> > **Response to the Authors' Rebuttal**
> >
> > I am very glad to hear that the authors are considering to introduce some empirical experiments in the future. The extension to general graphs are also highly expected. Looking forward to seeing these future work. Thanks!

---

> > > ### Author Response · Authors · 2023-08-16
> > >
> > > Thank you for the response and we are glad to hear that you share our excitement for this line of research! To be clear, we think that future work on empirical experiments or general graphs may benefit from our submission, but are outside its scope. Thanks!

---

### Official Review · Reviewer_Craf · 2023-07-18

**Soundness:** 4 excellent
**Presentation:** 4 excellent
**Contribution:** 3 good
**Rating:** 6
**Confidence:** 4

**Summary:**

Given a graph $G$, the effective resistance approximation problem is one where you are asked to approximate the effective resistances of some subset $S$ of vertex pairs. The authors have two contributions:

Firstly, the authors demonstrate an $O(m\epsilon^{-1} + |S|)$ algorithm for computing effective resistances in graphs with constant expansion. This matches the current best known algorithm for computing effective resistances in expanders for dense graphs, and beats it in sparse graphs of total edge count $o(n\epsilon^{-2})$. To achieve this, the authors make use of CountSketch, and demonstrate that there are sets of vectors from which the effective resistances can be extracted in $\tilde{O}(1)$ time. These vectors are obtained through the use of CountSketch on vectors with small $\ell_1$ norm. Furthermore, this data structure is $\tilde{O}(n\epsilon^{-1})$ sparse.

Secondly, the authors demonstrate an $\Omega(n^2\epsilon^{-1/2})$ conditional lower bound on the time required to compute the effective resistance on all pairs of edges in a graph. They demonstrate that if an effective resistance approximation algorithm is faster than $O(n^2\epsilon^{-1/2})$ time by a polynomial factor, then the time complexity of triangle detection will be truly subcubic. Currently, the only subcubic algorithms for triangle detection require fast matrix multiplication, so it is unlikely that a combinatorial algorithm for the effective resistance problem on all pairs can be faster. Furthermore, they improve on the lower bounds on a range of similar spectral sum problems using this same reduction, by refining the analysis performed by [4].

They achieve this improvement through the use of several techniques. Firstly, by considering a different power series expansion as in [4], they demonstrate that the triangle counting problem can be reduced to a different problem that looks more like effective resistances. To obtain sufficiently good estimation errors, they also make use of symmetric random signing, where each entry of the adjacency matrix is flipped with probability 1/2.

**Strengths:**

Originality:

I really like the approaches used by the authors. I think that the technical contributions here in both the upper and lower bounds are very interesting to me.

I like the use of the asymmetric CountSketch, and the fact that the vectors picked are small in $\ell_1$ norm. The usage of a different power series expansion to obtain a different reduction for the triangle detection problem is also very interesting to me. The former is novel to me, and the latter reads like a novel improvement to obtain better bounds than a preexisting argument.

Quality/Clarity:
The full paper is a bit of a rough read, which is the nature of theory papers, but I like that a lot of the intuition is explained in the main section, that makes it a lot easier to understand what is happening. I could follow/guess a lot of what was going to happen in the analysis just from reading the main section which had none of the analysis.

Significance:
The upper bound is relatively significant. Effective resistance approximation is certainly a very important problem that is used as a subroutine for many other problems. They have made an improvement to the approximation time complexity in expanders in sparse graphs, which is a significant contribution to this area. In terms of lower bounds, they have made a significant refinement of the previous approach, which results in an improvement on the lower bounds for a large number of different problems.



**Weaknesses:**

The primary weakness of the upper bound is that the contribution made here is relatively marginal in scope. The paper has made improvements in solving a problem that is important, but have only done so in a limited setting (sparse expanders).

The lower bound is also a very interesting refinement of the previous approach, but it is somewhat unclear to me why this lower bound specifically is one that is important to look at. I feel like there are several other natural lower bounds for effective resistances that are more natural to want to consider. This latter point might just be me since I am not very well versed in this area, but even though this reduction to triangle detection without the use of FMM seems pretty standard is used in many papers, and I understand the arguments as to why it is done, I can't really find myself getting behind it.

**Questions:**

The lower bound obtained by the authors is one on the time required to compute the effective resistance on all pairs of edges. I wonder if there are any lower bounds on the time required to compute the effective resistance on just the edges of the graph. Looking through the use cases of effective resistances in the introduction. It seems to me that a lot of them only care about the effective resistance of just the edges in the graph, while maybe only the most recent works on over-squashing in GNNs care about the total effective resistance.

I am also very confused about the implications of the lower bound with regards to the upper bound. The lower bound is $\Omega(n^2\epsilon^{-1/2})$, while the upper bound is $O(m + n\epsilon^{-2} + n^2)$. Even if the graph were not sparse, the algorithm takes $O(n\epsilon^{-2} + n^2)$ time, while the lower bound takes $\Omega(n^2\epsilon^{-1/2})$. If we pick $\epsilon$ to be small, say $\sqrt{n}^{-1}$, isn't the upper bound faster than the lower bound? I don't think I understand lines 82-84, where the authors claim that this lower bound holds on expanders.

**Limitations:**

Typos:

Line 240: peridicular
Line 521: resistance
Line 536: reisistance

---

> ### Author Rebuttal · Authors · 2023-08-09
>
> Thanks for your thoughtful comments and questions! We discuss more motivation for considering the expander case in the general response.
>
> **(1) Why consider conditional lower bounds on algorithms using the subcubic non-FMM triangle detection hardness assumption? This is standard, but I can’t find myself getting behind it.**
>
> It’s an interesting question to obtain other lower bounds, such as interesting lower bounds on tradeoffs between the query time versus sketch size for effective resistance sketching. However, effective resistance estimation and sketching already have various polynomial time algorithms, and unconditional polynomial lower bounds for explicitly polynomial time solvable problems have been elusive. So, it isn’t clear how to approach unconditional runtime lower bounds.
>
> We wanted to offer some additional reasoning for why the conditional lower bound (based on subcubic non-FMM triangle detection) considered in our paper and [4] is useful _for algorithm designers_. Several recent advances in effective resistance estimation algorithms (in particular, [1], [2], and [3]) don't use FMM. Our lower bounds provide the valuable insight that _FMM is likely essential_ to cross a certain runtime threshold. We show that  to obtain better than $\tilde{O}(n^2\epsilon^{-1/2})$ runtime for effective resistance estimation, an algorithm designer must either (1) change the approach of existing methods to figure out _how to incorporate FMM_ into the algorithm (and people debate the practical utility of FMM-based algorithms); or (2) achieve the _algorithmic breakthrough_ of designing subcubic triangle detection algorithms without FMM. Triangle-detection-based reductions highlight that this particular difficulty is also a barrier for several other problems.
>
> **(2) Are there lower bounds for the problem of estimating effective resistance of just all edges?**
>
> Yes. Our lower bound $\tilde{\Omega}(n^2\epsilon^{-1/2})$ time directly implies an $\tilde{\Omega}(m\epsilon^{-1/2})$ time lower bound on effective resistance estimation for all edges in a general expander graph. We provide context in the next paragraph and then provide the technical reduction in the third paragraph below.
>
> Your question is a good one, and it gets at the fact that one could ask the following two interesting questions relating to effective resistance estimation of all edges in a graph:
>
> 1. **Our work answers this:** Barring a breakthrough in combinatorial (i.e., non-FMM) algorithms, can we develop a combinatorial algorithm which takes as input an **arbitrary** expander graph on $n$ nodes and $m$ edges and outputs estimates of effective resistances of all $m$ edges in faster than $\tilde{\Omega}({m\epsilon^{-1/2}})$ time? **Our results say NO; we justify technically below.**
>
> 2. **Our work doesn't answer this:** Barring a breakthrough in combinatorial algorithms, can we develop a combinatorial algorithm which takes as input a **sparse** expander graph on $n$ nodes and $m < f(n)$ edges and outputs estimates of all $m$ edges in $o(T)$ time?  **Our results leave this question open**, and we are not aware of any lower bounds which answer this question for any nontrivial $T$ when $f(n) = o(n^2)$. This question is more open-ended, because one can choose what sparsity $f(n)$ we demand of the input graph, and it's unclear what choice of $f$ and $T$ would lead to interesting results. We are curious about it as a question future work.
>
> **Technical justification for 1:** Let $1$ be the ones vector. The idea is, given $n$-node, $m$-edge graph $G$, we can add a complete graph of edges of sufficiently small weight that would not change the effective resistances much. Let $L_G = D-A$ be the graph Laplacian and $H$ be the graph obtained by adding a complete graph with uniform edge weight $\alpha > 0.$ Then, $L_H = D - A + \alpha (nI - 11^T)$. Let $M = L_H +  \alpha 11^T = D - A + \alpha n I$. As $M$ just completes the kernel of $L_H$, $x^T M^{-1}x = x^\top (L_H)^\dagger x$ for all $x \perp 1$. And, $M^{-1} = (D - A + \alpha (nI))^{-1} = (Q(\Lambda + \alpha nI)^{-1} Q^\top$ where $Q\Lambda Q^\top$ is the eigen-decomposition of $D-A$. The entries $((\Lambda + \alpha nI)^{-1})_i$ can be made arbitrarily close to $(\Lambda^{-1})_i$ by taking $\alpha$ small enough. So, $x^\top {L_H}^{\dagger} x = x^\top M^{-1} x$ can be made arbitrarily close to $x^\top {L_G}^{\dagger} x$ for any $x \perp 1$. So we can get approximate **all pairs** effective resistances on $G$ from approximate **all edges** effective resistances on $H$. So, any combinatorial algorithm faster than $\tilde{\Omega}(m\epsilon^{-1/2})$ for the all-edges problem would immediately imply a combinatorial algorithm faster than $\tilde{\Omega}(n^2\epsilon^{-1/2})$ for the all-pairs problem– the latter is conditionally ruled out by Theorem 3. We may include this discussion in the next iteration.
>
> **(3)  [3] obtains an $\tilde{O}(n^2 + n\epsilon^{-2})$ runtime, which is better that the lower bound if $\epsilon = 1/\sqrt{n}$. Is this a contradiction?**
>
> No, there is no contradiction. When we claim a conditional lower bound of $\tilde{\Omega}(n^2 \epsilon^{-1/2})$, we only mean that it is (conditionally) impossible to obtain a combinatorial all pairs effective resistance algorithm that runs in less than $\tilde{O}(n^2\epsilon^{-1/2})$ time for all $n$ and $\epsilon$ regimes. This doesn’t rule out the fact that there may exist algorithms which run faster than $\tilde{O}(n^2\epsilon^{-1/2})$ for certain $\epsilon$ regimes. The upper bound of $\tilde{O}(n\epsilon^{-2} + n^2)$ from [3] is only better than $\tilde{O}(n^2\epsilon^{-1/2})$ for certain regimes of $\epsilon$. In the proof of our lower bound we choose $\epsilon = \tilde{O}(n^{-2})$ (see lines 825-830 in the supplemental material). This implies that it is (conditionally) impossible to obtain a combinatorial algorithm that runs in less than $\tilde{O}(n^2\epsilon^{-1/2})$ time **for all** $n$ and $\epsilon$ regimes.

---

> > ### Comment · Reviewer_Craf · 2023-08-11
> >
> > This clarifies a lot of the questions I had, thank you very much!

---

### Official Review · Reviewer_T2X2 · 2023-07-25

**Soundness:** 4 excellent
**Presentation:** 3 good
**Contribution:** 3 good
**Rating:** 6
**Confidence:** 3

**Summary:**

The paper studies the problem of estimating the effective resistances in $n$-node $m$-edge undirected, expander graphs, and provides a $O(m/\epsilon)$ time algorithm which produces an effective resistances sketch in $O(n/\epsilon)$ size of space. This improves the runtime of the previous algorithm that estimates the effective resistance of all edges in such graphs and can be extended to the general problems of sketching the pseudoinverse of positive semidefinite matrices and estimating functions of their eigenvalues. The authors also give a new corresponding conditional lower bound where the $\epsilon$ dependence is improved.

**Strengths:**

1. The technical contribution is solid. The paper provides the algorithms with better theoretical bounds, along with the corresponding lower bound. To achieve this, one key point of the paper is it considers an asymmetric factorization and the asymmetric CountSketch in $\ell_1$.

2. The organization of this paper is good. The introduction section is detailed and the authors then first give a technical overview which can help the readers first get a brief understanding of the key point in the algorithm design.

**Weaknesses:**

1. This paper is a pure theory paper. However, given that it is a NeurIPS submission, I think the experimental evaluations will make the paper stronger.

2. I feel like some parts of the paper are a little technically dense and not very easy to follow. Also, the paper uses many standard facts in spectral graph theory. I think it will make the presentation more clear if the authors can give more detail in the preliminaries section.



**Questions:**

1.As summarized in table 1,  [3] has a runtime $O(m + n/\epsilon^2)$, will this bound be better than this work if $m/n > 1/\epsilon$?

**Limitations:**

See the above questions.

---

> ### Author Rebuttal · Authors · 2023-08-09
>
> Thank you for your thoughtful comments and questions! We have discussed interesting directions for future empirical experiments, as well as the modifications to the preliminaries section which you suggested in the general response. Please let us know if you have any additional feedback on those points.
>
> **Question 1: Is the result of [3] better if $m/n > 1/\epsilon$.**
>
> **Response:** Yes, but we can almost match it. In the regime where $m/n >> 1/\epsilon$, [3] yields a better runtime than our $\tilde{O}(m \epsilon^{-1})$ result in this regime. We can almost match this runtime by noting that our algorithm also immediately yields an $\tilde{O}({m^{1+o(1)} + n \epsilon^{-2}})$ runtime. This argument is discussed in Line 118-124 as well as in the caption of Table 1/Table 2; however we summarize the argument briefly in bullet 1 below. Consequently, in the dense regime where $m/n >> 1/\epsilon$, the result of [3] is better than ours only by an $\tilde{O}(m^{o(1)})$ factor. As our runtime was obtained independently of [3], we did not consider whether it is possible to remove the $m^{o(1)}$ factor gap between our result and that of [3] in this regime, but it is an interesting question for future work. We point to one approach for trying to remove this gap in bullet 2 below.
>
> * **Summary of procedure to match [3] in dense regime $m/n >> 1/\epsilon$**: Given $G$, in $\tilde{O}(m^{1+o(1)})$ time, we can use the result of [1] to construct a sparse “sketch” graph $H$ such that $H$ has only $n \epsilon^{-1}$ edges and $r_H(i,j) \approx_{\epsilon/4} r_G(i,j)$ for all nodes $i,j \in V$. Now, since $H$ is sparse, in only $\tilde{O}(n\epsilon^{-2})$ additional time, we can run our effective resistance estimation algorithm on $H$ to recover $(1 \pm \epsilon)$ multiplicative approximations to $r_G(i,j)$.
>
> * **One approach towards removing the $m^{o(1)}$ factor gap in dense regime $m/n >> 1/\epsilon$**: One approach towards removing the $m^{o(1)}$ factor gap between our results and [3] would be to improve the results of [1] to design an algorithm that generates the sparse “sketch” graph $H$ described above using only $\tilde{O}(m)$ time (instead of $\tilde{O}(m^{1 + o(1)}$). If such an algorithm could be obtained, then the procedure discussed in bullet 1 above would immediately run in $\tilde{O}(m + n\epsilon^{-2})$ time, allowing us to remove the $m^{o(1)}$ factor gap and match the result of [3] in the regime where $m/n >> 1/\epsilon$.

---

> > ### Comment · Reviewer_T2X2 · 2023-08-18
> >
> > Thanks for the detailed response, it addresses most of my concerns!

---

### Author Rebuttal · Authors · 2023-08-09

We thank the reviewers for their helpful comments and questions. We are encouraged that the reviewers appreciated novelty in our technical contributions (R2, R3, R5) and cleverness in our techniques for obtaining our improved upper and lower bounds (R4, R6). We are glad that they found our paper to be well-organized/well-written (R1, R3, R4) and our discussion of the approach to be helpful for providing intuition for the proofs (R2, R5). We are also glad that reviewers recognized the motivation for studying the effective resistance estimation problem (R3, R4).

In this general response, we will respond to some common comments/questions relating to the submission. Additional reviewer-specific questions are answered in the individual responses.

**Comment 1: Additional content to incorporate in the Intro/Prelims}** (R1, R5). Based on the reviewers’ suggestions, below is a list of a changes we plan to make to make to clarify the notation and our results in next iteration of our submission:
* Define the edge incidence matrix in the prelims (R5).
* Include more background on standard facts in spectral graph theory in the preliminaries (e.g., Cheeger’s inequality, equivalent expressions for effective resistance, etc.) (R1, R5).
* Move the informal discussion of the triangle detection problem definition and hardness result from the introduction into a definition/conjecture environment for increased clarity (R5).
* Define $\tilde{\Omega}$ clearly in the prelims (Section 2) and add a forward reference when $\tilde{\Omega}$ and $\tilde{O}$ is first mentioned in Section 1 (R5).
* Define the Johnson-Lindenstrauss (JL) projection matrix (R5).

We welcome any other suggestions the reviewers may have for additional background that would be helpful to incorporate into the introduction/prelim of the paper.

**Comment 2: Empirical validation** (R1, R3). R1 and R3 commented that empirical validation of our algorithm against other effective resistance estimation algorithms would be interesting. We agree an extensive empirical comparison of various effective resistance estimation algorithms would be an intriguing direction for future work. One potential obstacle in this direction is that we are unaware of the extent to which previous algorithms for effective resistance estimation such as that of [1], [2], and [3] have been implemented. Nevertheless, we believe it would be interesting to do a comprehensive comparison of the variety of effective resistance estimation algorithms on synthetic and real-world datasets, however this is outside the scope of our current work.

**Comment 3: What are the bottlenecks in extending the algorithmic upper bound of $\tilde{O}(m\epsilon^{-1})$ to general graphs, and what is the motivation for considering expanders?** (R3, R6, and generally touched on by all reviewers). It is indeed a limitation of our analysis that we are only able to shown an $\tilde{O}(m\epsilon^{-1})$ algorithm for expanders, and it is an interesting open question to obtain an $\tilde{O}(m\epsilon^{-1})$ effective resistance estimation algorithm for general graphs. As we have discussed and R6 mentions, in several spectral graph theoretic problems (such as spectral sketching, sparsification, etc.) a key step towards faster algorithms on general graphs is to obtain faster algorithms on expanders. In a variety of problems studied in the literature, fast algorithms on expanders have been combined, for example, with expander decomposition algorithms to generalize algorithms on expanders to algorithms which can obtain comparable guarantees on general graphs. Consequently, the effective resistance estimation problem on expanders is an interesting previously studied problem in the literature ([3, 33]).

While our techniques do not yield an $\tilde{O}(m\epsilon^{-1})$ time algorithm for general graphs, we have shown our algorithm itself can still be applied to general graphs and may give some interesting results for graphs whose conditioning does not grow too quickly with m or n. For general graphs, the only difference is that the runtime becomes $\tilde{O}(m\bar{\kappa}(G)\epsilon^{-1})$, where $\bar{\kappa}(L_G)$ is the normalized pseudo-condition number of the graph, which we have defined in the preliminaries. For expanders, $\bar{\kappa}(G) = \tilde{O}(1)$, enabling our main result. However, we believe our result still gives an interesting runtime bound for families of graphs where $\bar{\kappa}$ does not grow too quickly with $n$. For instance, compared to the results $\tilde{O}(m\epsilon^{-1.5})$ of [1] our result gives an interesting tradeoff between $\bar{\kappa(L_G)}$ and $\epsilon$. Likewise, compared to the result $\tilde{O}(m + n\epsilon^{-2}\bar{\kappa}^3(L_G))$ obtained by [3], our result is better when $\bar{\kappa}(L_G) > (m \epsilon/n )^{1/2}$.

The main bottleneck in extending our results to the general graph case is that for general graphs, we do not know how to bound the l1 norm of the second term in the asymmetric factorization of effective resistance  to be $\tilde{O}(1)$ (this was also well-articulated by R6). To be mathematically concrete, we do not have a strong enough lower bound on the second eigenvalue of the normalized graph Laplacian (or equivalently, a strong enough upper bound on the normalized pseudo-condition number of the graph) in order to ensure that the power series expansion converges quickly in l1 norm. For general graphs, we were only able only bound the l1 norm of the second term in the asymmetric factorization to be $\tilde{O}(\bar{\kappa}(G))$, leading to the $\tilde{O}(\bar{\kappa}(G) m \epsilon^{-1})$ time algorithm for general graphs. Compared to the work of [3], our algorithms have a better dependence on $\bar{\kappa}(L_G)$ for general graphs. Technical details of this result are explained in more detail in Section 6.1 and Section 6.2.3 in the supplementary.

---

### Decision · Program_Chairs · 2023-09-21

**Decision:**

Accept (poster)

**Comment:**

This paper received universally strong reviews, and all were in agreement that it makes an interesting theoretical contribution to an important problem (effective resistance estimation in sparse graphs), using novel techniques. The algorithmic contribution is limited in scope as it applies specifically to expander graphs. However, this is an interesting case which has been studied in prior work.

We encourage the authors to take reviewer suggestions on presentation into account when revising the paper. While perhaps beyond the scope of this paper, we also encourage them to consider an empirical evaluation of their work and related baselines.